# In-vivo programmable acoustic manipulation of genetically engineered bacteria

Ye Yang [1,2,4], Yaozhang Yang[1,3,4], Dingyuan Liu[1], Yuanyuan Wang[1], Minqiao Lu[1], Qi Zhang [1], Jiqing Huang[1], Yongchuan Li[1], Teng Ma[1,2] ✉, Fei Yan [1,2] ✉ & Hairong Zheng [1,2] ✉

Acoustic tweezers can control target movement through the momentum interaction between an acoustic wave and an object. This technology has advantages over optical tweezers for in-vivo cell manipulation due to its high tissue penetrability and strong acoustic radiation force. However, normal cells are difficult to acoustically manipulate because of their small size and the similarity between their acoustic impedance and that of the medium. In this study, we use the heterologous expression of gene clusters to generate genetically engineered bacteria that can produce numerous sub-micron gas vesicles in the bacterial cytoplasm. We show that the presence of the gas vesicles significantly enhances the acoustic sensitivity of the engineering bacteria, which can be manipulated by ultrasound. We find that by employing phased-array-based acoustic tweezers, the engineering bacteria can be trapped into clusters and manipulated in vitro and in vivo via electronically steered acoustic beams, enabling the counter flow or on-demand flow of these bacteria in the vasculature of live mice. Furthermore, we demonstrate that the aggregation efficiency of engineering bacteria in a tumour is improved by utilizing this technology. This study provides a platform for the in-vivo manipulation of live cells, which will promote the progress of cell-based biomedical applications.

Cell-based therapy has emerged as a novel strategy for treating various diseases, such as tumours, myocardial necrosis, bone diseases, and even brain disorders[1,2]. Live cells, including CAR-T cells[3], stem cells[4] and oncolytic bacteria[5], have been widely investigated for use in disease therapy, and some have exhibited promising application prospects. There are two main routes for administering these cells. One involves directly transplanting these live cells into the disease site (non-systemic administration)[6], and the other involves injecting them into the circulatory system (systemic administration)[7]. Non-systemic administration is a suitable therapeutic route when treating diseases with definite locations because it produces a therapeutic effect locally at the application site. However, the distribution of therapeutic cells via this route is very limited compared to that in systemic

administration. Systemic administration is the optimal therapeutic route when treating disseminated ailments because it distributes the cells throughout the body, including at the action site[8,9]. Consequently, systemic administration of live isolated cells or genetically engineered cells, such as bone marrow stromal cells, CAR-T cells, and even genetically engineered bacteria, has become an attractive strategy for various pre-clinical and clinical studies[10–12]. These cells, once delivered by a systemic route, migrate through the systemic circulation, resettle at the damaged sites, and receive local signals that execute functions or direct tissue differentiation. Numerous articles have demonstrated that the highly efficient migration and resettlement of these therapeutic cells to the damaged sites is essential for their efficacy[13,14]. Therefore, it is desirable to develop a strategy to manipulate these

[1]Shenzhen Institutes of Advanced Technology, Chinese Academy of Sciences, 518055 Shenzhen, China. [2]Shenzhen College of Advanced Technology, University of the Chinese Academy of Sciences, 100049 Beijing, China. [3]Shenzhen Bay Laboratory, 518132 Shenzhen, China. [4]These authors contributed equally: Ye Yang, Yaozhang Yang. ✉e-mail: teng.ma@siat.ac.cn; fei.yan@siat.ac.cn; hr.zheng@siat.ac.cn

systemically delivered cells to migrate them with high efficiency to the diseased sites.

To date, several technologies have been developed for the manipulation of particles or cells, including optical[15,16], magnetic[17,18] and acoustic tweezers[19,20]. Among these technologies, optical tweezers can use highly focused laser beams to trap and manipulate microscopic objects in a non-contact manner through photon momentum transfer. This technology is widely used for precise cell manipulation and has led to cutting-edge research, such as studies of the deformation mechanism of human red blood cells[21,22], mechanochemical processes in cells[23], and shear modulus of the human erythrocyte membrane[24]. However, optical tweezers can only be used in transparent media, which greatly limits their in vivo applications. In addition, direct irradiation of high-intensity lasers on targets may cause photodamage to biological samples and photothermal or photochemical inactivation of bioactive drugs. Magnetic fields can pass through non-transparent media, enabling the use of magnetic tweezers to manipulate cells in vivo[25]. The targets that are exposed to the applied magnetic field need to be magnetised or connected to magnetised particles[26,27]. Unfortunately, magnetic tweezers have difficulty accurately locating the target position in vivo due to the unfocused nature of the magnetic field. Occasionally, magnetic labelling will have an irreversible effect on the viability and bioactivity of biological samples. Intriguingly, acoustic tweezers have recently emerged as promising tools for biological particle manipulation due to their low damage, high tissue penetrability, relatively high spatial precision on the micron scale and ability to be applied in various media[28,29]. An acoustic wave is a typical mechanical wave carrying acoustic energy and momentum and will be absorbed, scattered, and reflected when encountering particles. The acoustic radiation force (ARF), which is generated by the exchange of momentum and energy between an acoustic wave and particles, enables ultrasound to manipulate objects whose densities or compressibilities differ from those of their surrounding media. Acoustic manipulation of cells has already been achieved in vitro, including in on-chip cell and organism engineering in microfluidic channels[30–32] and single-beam manipulation for cell characterisation and stimulation in liquids[33–35].

Some research has been published on in vivo manipulation of non-living objects, such as the manipulation of glass spheres in a pig bladder[36], the trapping of microbubbles in mouse back epidermal blood vessels[37] and the manipulation of microparticles in zebrafish embryos[38]. However, the in vivo acoustic manipulation of cells has not yet been reported. Two main challenges exist in this regard. First, the ARF is proportional to the third power of the cell radius, causing the force exerted on the biological cells (typically only a few microns) to be too weak to manipulate them[39,40]. Second, the acoustic impedance of the cells (typical value is between 1.6 and $1.8 \times 10^6$ rayls) is very close to that of the medium (for example, the values of water and soft tissue average are $1.5 \times 10^6$ and $1.58 \times 10^6$ rayls, respectively)[41,42], resulting in less ARF generation. Therefore, it is feasible to increase the acoustic impedance mismatch between the cells and medium to enhance the ARF exerted on the cells, resulting in possible acoustic manipulation at lower frequency and intensity.

Recently, synthetic biological technology has emerged as a powerful tool in cell engineering. By using this technology, many submicron gas vesicles (GVs, which are usually approximately rod-shaped structures with a length of 100–600 nm and a width of 40–200 nm) have been successfully generated in E. coli and HEK 293T mammalian cells, enabling their imaging[43,44] and in vitro manipulated[45] by ultrasound.

In this study, we aim to realise the acoustic manipulation of GV-expressing bacteria by using the holographic acoustic tweezers technology in vivo and further validate the therapeutic potential of our approach for tumours. In detail, we genetically engineered E. coli through the heterologous expression of gene clusters encoding GVs,

which significantly enhanced the acoustic contrast of these bacteria relative to water (Fig. 1, the acoustic contrast has increased by nearly 15 times), consistent with previous results[45]. We also designed and fabricated a 3 MHz 64-element ($8 \times 8$) 2D matrix array transducer for acoustic manipulation of these genetically engineered bacteria. Using this transducer, the acoustic waves can be focused to produce a strong ARF, trapping the genetically engineered bacteria (called GVs@E. coli) and driving them to move along the preset route. The maximum simulated ARF exerted on GVs@E. coli is about 63-fold higher than that of the control bacteria without GVs. Importantly, the GVs@E. coli, but not the control bacteria (E. coli), could be manipulated in the circulation by the programmable pulses after systemic administration, making them move reverse or flow on demand into the preset blood vessels. Moreover, we further demonstrated that trapping GVs@YB1 at the tumour site using holographic acoustic tweezers can significantly improve the migration and resettlement of bacteria at the tumour site and effectively slow down the growth rate of the tumour. Thus, this in vivo cell acoustic manipulation technology, based on the combination of biosynthetic GVs and acoustic tweezers, provides a method of directionally driving therapeutic cells for various biomedical applications.

## Results

### ARF-dominated acoustic trapping of GVs@E. coli

Using genetic engineering techniques (see the Methods section for details), a GV gene cluster combining the structural gvpA and gvpC genes from A. flos-aquae with the accessory genes gvpR-gvpU from B. megaterium was transformed into the E. coli BL21 (A1). GVs were generated in the engineered bacteria when they were exposed to 0.5% L-arabinose and 0.4 mM isopropyl β-D-thiogalactopyranoside (IPTG) inducers, enabling them to respond to acoustic beams and be aggregated by the ARF (Fig. 2a and Supplementary Fig. 1). As expected, no GVs could be observed in the genetically engineered but uninduced E. coli BL21 (control E. coli) under phase-contrast microscopy, whereas numerous GVs could be observed in the genetically engineered cells when they were activated by chemical inducers (Fig. 2b, d). Transmission electron microscopy (TEM) further confirmed that the GVs were successfully expressed in the genetically engineered GVs@E. coli, with round or oval nanoscale structures occupying the cytoplasm of the bacteria, whereas there was no similar structure in the uninduced E. coli BL21 (Fig. 2c, e).

The acoustic tweezers were designed based on a self-designed 3 MHz, 64-element ($8 \times 8$), 2D matrix array. As shown in the simulated 3D acoustic field (Fig. 2f), a tiny focal beam (focal depth: 6 mm, f-number = 1) could be generated by exciting the array with an iterative backpropagation (IB) algorithm[46]. The focal zone, which was identified by a peak acoustic pressure of −3 dB, was ~0.85 mm (1.7 $\lambda$) and 4.1 mm (8.2 $\lambda$) in the lateral and axial directions, respectively (Supplementary Fig. 2a–f). The maximum peak-to-peak acoustic pressure in the focal zone was measured to be approximately 1.68 MPa (excitation voltage: 15 V). To validate the acoustic manipulation of bacteria in our system, we first examined the trapping capabilities of genetically engineered GVs@E. coli and uninduced control E. coli by using the acoustic tweezers. Because the difference between the GVs@E. coli and control bacteria lies only in the presence or absence of GVs, it can contribute to the fact that the abundant cytoplasmic GVs significantly decrease the average density and increase the compressibility, resulting in the acoustic contrast of the genetically engineered bacteria changing from +0.07 to −1.1, flipping the sign of the acoustic contrast from positive to negative, and increasing its magnitude by approximately 15 times[45]. Thus, the control bacteria without GVs, which possess positive acoustic contrast, are directed away from the focus by the ARF (Fig. 2g) and are too weak to overcome the Stokes force caused by the acoustic streaming[47]. By contrast, the negative acoustic contrast of the engineered bacteria makes the direction of the ARF exerted on them point

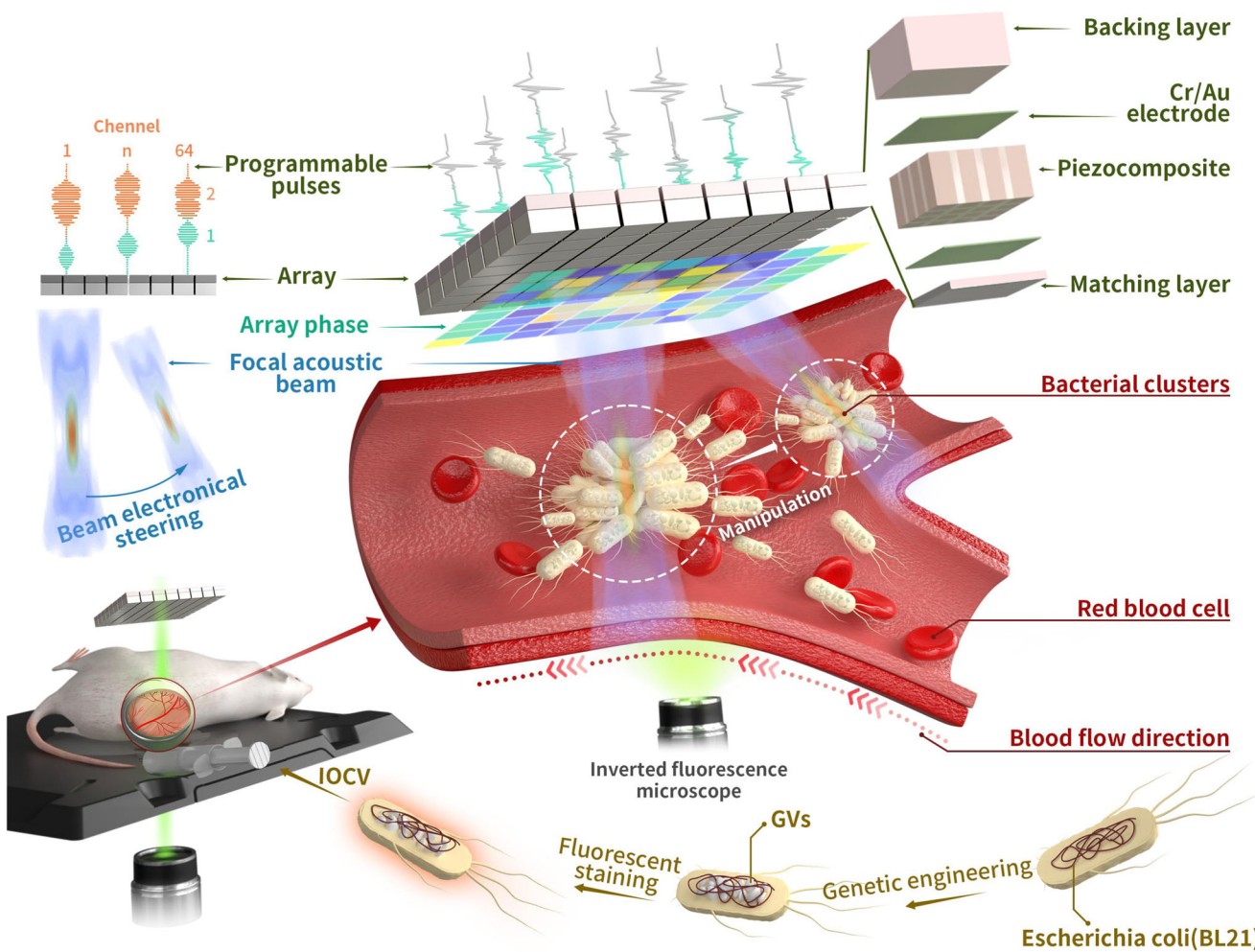

**Fig. 1 | Schematic of in vivo acoustic manipulation of genetically engineered bacteria.** The *E. coli* BL21 was genetically engineered by introducing a gene cluster encoding GVs, endowing these bacteria (GVs@*E. coli*) with distinct acoustic impedance relative to water. When these GVs@*E. coli* were systemically administrated into mice, they could be trapped into clusters and programmatically manipulated by acoustic tweezers equipped with a 3 MHz 64-element (8 × 8) 2D matrix array transducer. By employing electronically controlled beam-steering techniques, multi-programmable acoustic beams could be generated by the phased-array 2D matrix transducer, achieving the counter flow or on-demand flow of these bacteria into the preset blood vessels in live mice. Using the fluorescence-labelled GVs@*E. coli*, the manipulation process could be clearly observed via the mouse dorsal skin-fold window chamber model under the inverted fluorescence microscopy.

to the centre of the focal field (Fig. 2h). The larger contrast magnitude due to the presence of GVs enables them to exert stronger ARF at the same input ultrasonic energy, and the maximum calculated ARF is about 63-fold higher than that of the control bacteria without GVs (Fig. 2i, see "Methods" for details). This strong ARF with the direction pointing to the focal centre will help these genetically engineered bacteria with GVs to overcome the Stokes force and will drive them to aggregate in the focal area[45]. Moreover, when multiple GVs@*E. coli* are close to the focal centre, cross-scattering of these reflections from GVs generates a localised pressure gradient and gives rise to secondary radiation forces which attract adjacent bacteria (mainly caused by the in-phase oscillation of GVs), and facilitate the GVs@*E. coli* to form a cluster[41]. Subsequently, the analyses of ARF exerted on the GVs@*E. coli* and control *E. coli* were further verified experimentally. When these two types of bacteria (labelled by fluorescent DiI) were separately injected into 500-μm-diameter silicone tubes, the GVs@*E. coli* were quickly aggregated at the focal centre (the bacteria cluster diameter was approximately 223 μm after 30 s ultrasound irradiation at 15 V excitation voltage), whereas the control bacteria, which did not have GVs, were not aggregated under the same focal acoustic field (Fig. 2j, k and Supplementary Movie 1). There are about $3-8 \times 10^5$ CFU in a typical

cluster of GVs@*E. coli* (Supplementary Fig. 3a, b), and each bacterium has about 200–500 GVs, forming about $1 \sim 3 \times 10^8$ GVs in a typical cluster of GVs@*E. coli*. And the GVs in the bacteria were not damaged by ultrasonic irradiation (Supplementary Fig. 3c).

Next, the bacterial trapping efficiency of the acoustic tweezers was investigated by changing the input voltage (5–30 V), duty cycle (1%–10%), and concentration of GVs@*E. coli* ($6.25-200 \times 10^7$ cells mL$^{-1}$). The trapping efficiency was assessed by calculating the size of the bacterial cluster after the ultrasound had been turned on for 5 min as well as the cluster formation time (150 μm cluster diameter) using fluorescence microscopy imaging. The results clearly showed that the cluster formation time greatly decreased from $137.4 \pm 10.9$ s to $3.8 \pm 0.3$ s as the input voltage increased from 5 to 30 V (duty cycle: 10%). When the input voltage was set above 15 V, a 150-μm-diameter bacterial cluster was generated within 15 s (Fig. 2l). Similarly, increasing the duty cycle from 1% to 7% (input voltage: 15 V) greatly decreased the cluster formation time from $208.0 \pm 34.1$ s to $16.2 \pm 8.5$ s. However, further increasing the duty cycle from 7 to 10% did not significantly decrease the formation time (Fig. 2m). In addition, we found that the final size of the bacterial cluster has little influence from the input voltage or duty cycle, but rather mainly depends on the bacterial

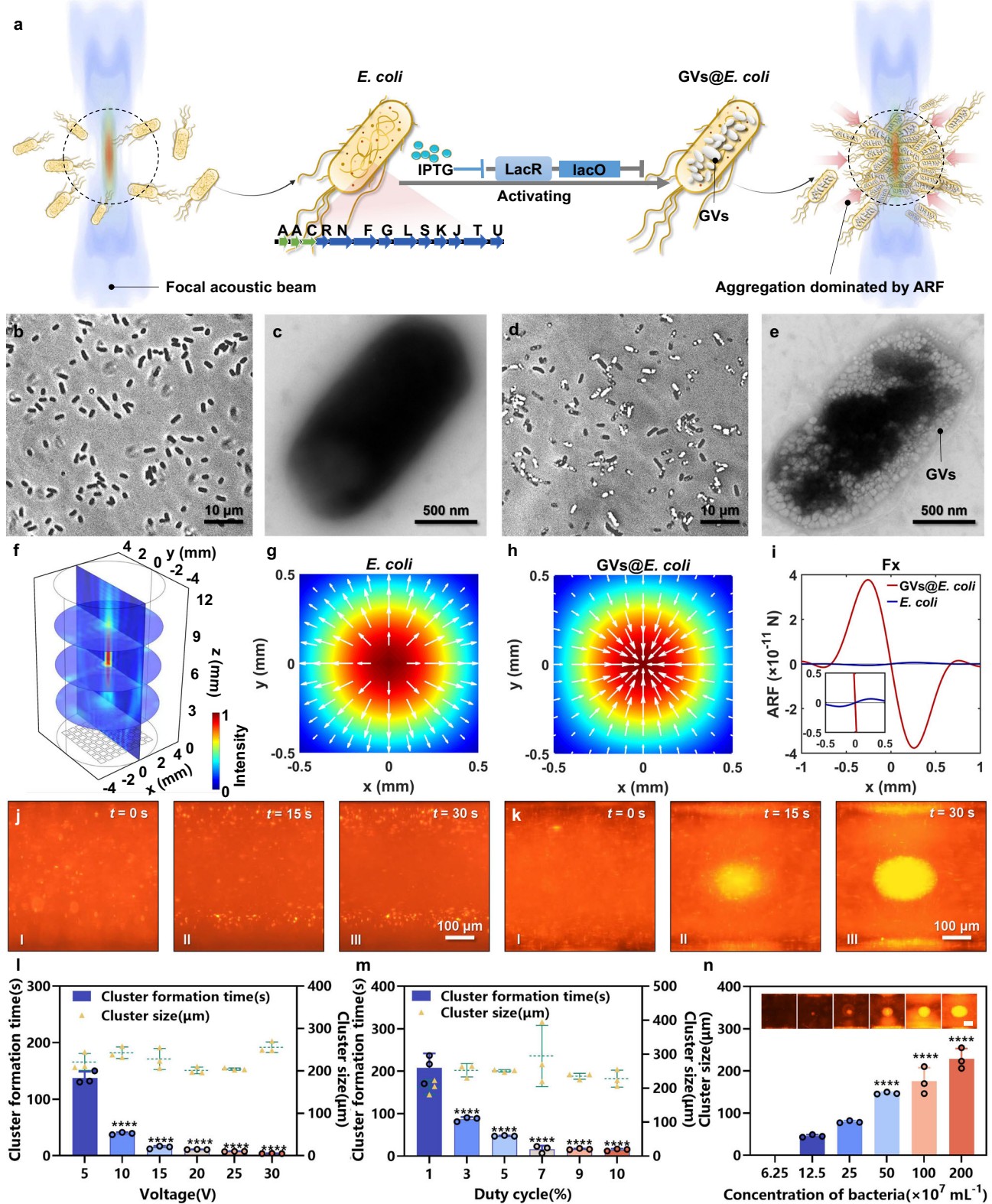

concentrations. The bacterial cluster is not generated at a concentration of $6.25 \times 10^7$ cells $mL^{-1}$, whereas the cluster size increases considerably from $45.6 \pm 3.1\,\mu m$ to $228 \pm 24.9\,\mu m$ when increasing the bacterial concentration from 12.5 to $200 \times 10^7$ cells $mL^{-1}$ (Fig. 2n). To maximise the trapping force generated by the acoustic tweezers and to avoid excessive peak negative pressure collapsing the GVs in the bacteria, the following three parameters were selected for subsequent in vitro experiments: 15 V input voltage, 10% duty cycle, and $200 \times 10^7$

cells $mL^{-1}$ bacterial concentration. The GVs in the bacteria will not collapse (Supplementary Fig. 4a) or cause inter-cell chemical reactions (Supplementary Fig. 4b) under these parameters.

**In vitro programmable acoustic manipulation of GVs@*E. coli***

As the array-based acoustic tweezers can produce complex acoustic fields that can be adjusted dynamically, they can perform complicated programmable manipulation processes[46,48]. To test this ability, we used

**Fig. 2 | ARF-dominated acoustic trapping of genetically engineered bacteria.**
**a** Schematic of the acoustic trapping of GVs@*E. coli*. **b** Phase-contrast microscopy
and **c** representative TEM images of uninduced control *E. coli*. **d** Phase-contrast
microscopy and **e** representative TEM images of GVs@*E. coli*. The black parts in
(**b**, **d**) are the cytoplasm of the bacteria, and the white areas in (**d**) represent the GVs.
Images are representative of three experiments in (**b–e**). **f** 3D simulated schematic
diagram of the focal acoustic field (focal point: (0, 0, 6) mm). Simulated acoustic
field (*x–y* plane, *z* = 6 mm) and ARF vectors of the **g** control *E. coli* and **h** GVs@*E. coli*
at the focal point, respectively. **i** ARF exerted on *E. coli* and GVs@*E. coli* by the focal
beam in the *x*–axis. Microscopic fluorescence photographs of **j** control *E. coli* and
**k** GVs@*E. coli* in a silicone tube being exposed to ultrasound (focal beam) for 30 s,

respectively. Only GVs@*E. coli* can aggregate at the focal beam centre and form
clusters under the dominance of the ARF. Bacterial cluster formation times and
cluster sizes of the GVs@*E. coli* under different **l** input voltages and **m** duty cycles.
**** means $P < 0.0001$ compared to the 5 V group in (**l**) or 1% duty cycle group in (**m**).
**n** Bacterial cluster size of the GVs@*E. coli* with various concentrations under the
same ultrasound conditions. **** means $P < 0.0001$ compared to the $12.5 \times 10^7$ cell
mL$^{-1}$ case in (**n**). Scale bar: 100 μm. Data in (**l–n**) are presented as mean ± s.d. ($n$ = 3
biologically independent samples per group). Statistical analysis was multiple
comparisons by using the one-way analysis of variance with Sidak's test. Source
data are provided as a Source Data file. The ARF-dominated acoustic trapping of
GVs@*E. coli* is shown in Supplementary Movie 1.

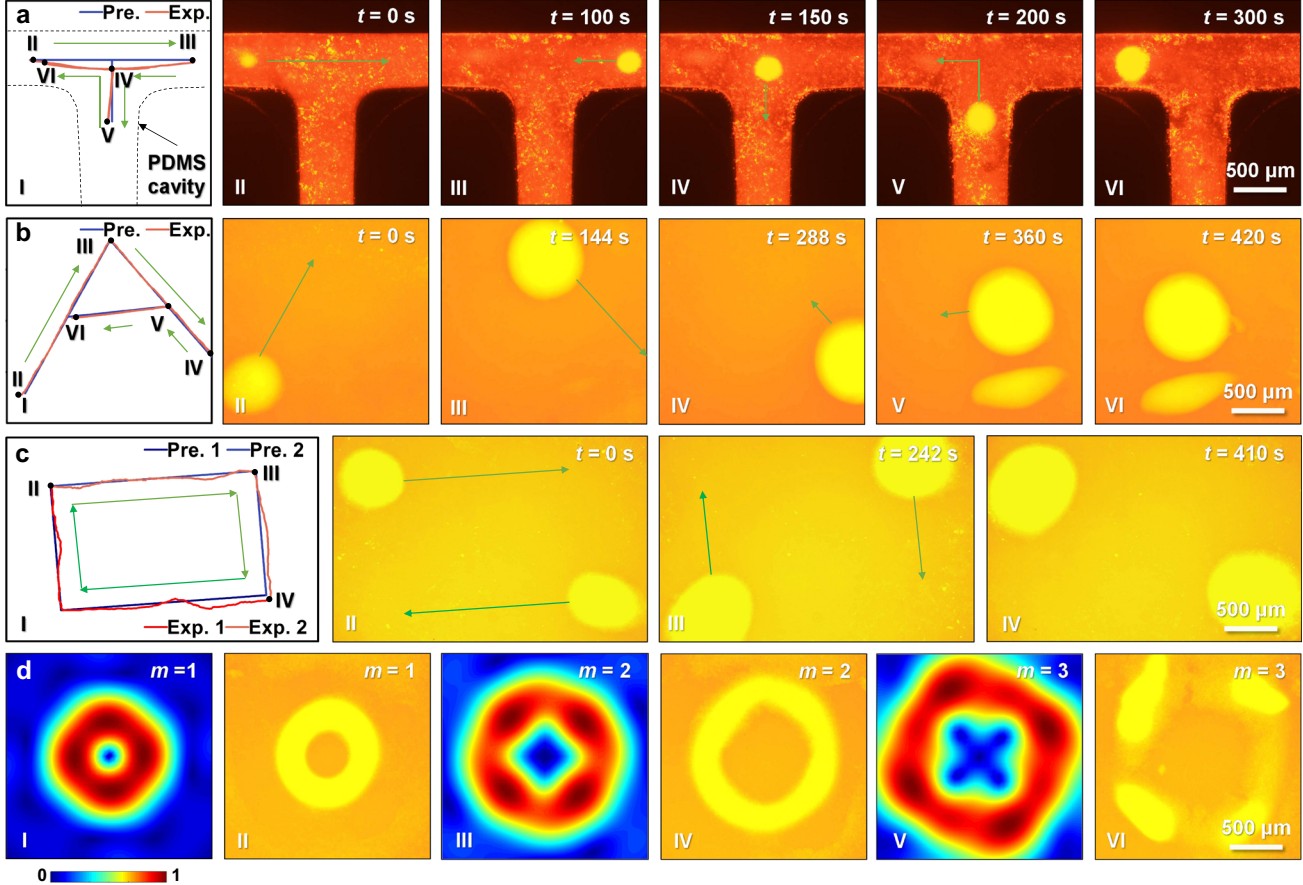

**Fig. 3 | In vitro programmable acoustic manipulation of GVs@*E. coli* cluster.**
**a** Manipulating the GVs@*E. coli* cluster selectively through the fork in a T-shaped
PDMS cavity by electronically steered focal beams. **b** GVs@*E. coli* cluster manip-
ulation along the A-shaped trajectory under boundary-free conditions (on the
bottom of an ordinary 10-cm-diameter Petri dish). I in (**a**, **b**): Preset and experi-
mental locomotion trajectories of the cluster. II–VI in (**a**, **b**): Time-lapse microscopic
fluorescence image sequences of the location of the manipulated GVs@*E. coli*
cluster. The green arrows indicate the translation directions. **c** Manipulation of two
GVs@*E. coli* clusters simultaneously, which start to move from the diagonal vertices
of the rectangle, then along the boundary of the rectangle toward the other

diagonal vertex of each. I: Preset and experimental locomotion trajectories of the
two clusters. II–IV: Time-lapse microscopic fluorescence image sequences of the
locations of the manipulated double GVs@*E. coli* clusters. The dark and light green
arrows represent the different directions of movement of the two clusters,
respectively. **d** Transformation of the vortex acoustic field patterns (where the
topological charge changed from $m = 1$ to $m = 3$) by using GVs@*E. coli*. I, III and V are
the vortex acoustic fields with $m = 1$, 2 and 3, respectively. II, IV and VI are micro-
scopic fluorescence images of the GVs@*E. coli* under the corresponding acoustic
fields in I, III and V. The in vitro programmable acoustic manipulation of the GVs@*E.
coli* cluster is shown in Supplementary Movie 1.

electronically steered focal beams to trap and drive these genetically
engineered GVs@*E. coli* along a programmable trajectory. As shown in
Fig. 3a and Supplementary Fig. 5a, and Supplementary Movie 1, the
bacterial cluster in a forked PDMS cavity can be manipulated selec-
tively through the forks. The entire trajectory of the cluster is displayed
in a "T" shape, matching the preset path perfectly. The average moving
velocity of the cluster is ~12 μm s$^{-1}$, also matching the programmed
20 μm movement every 2 s. Interestingly, only the bacteria inside the

focal zone could be trapped. Thus, as the focal zone moves in the
bacteria-filled cavity, new bacteria are continuously trapped, resulting
in a gradual increase in cluster size from 157 to 367 μm. This ability of
bacterial clusters to be driven programmatically along the pipes will
pave the way for acoustic manipulation in vessels of the living body.
In addition to the aggregation and manipulation of GVs@*E. coli* in
the cavities, we achieved these results under boundary-free conditions
as well. As shown in Fig. 3b and Supplementary Fig. 5b and

Supplementary Movie 1, we successfully manipulated the bacterial cluster along the "A"-shaped trajectory on the bottom of a 10-cm-diameter Petri dish, whose lateral size is much larger than the focal zone. Notably, the diameter of the bacterial cluster is closer to the size of the focal zone due to the lack of boundary limitations, and the cluster shape changes from oval to circle. In addition, the cluster size increases from 534 to 830 μm during movement.

Given that the array-based acoustic tweezers can produce multi-focal beams, we wondered whether multiple bacterial clusters could be manipulated along different trajectories simultaneously. As displayed in Fig. 3c and Supplementary Fig. 5c and Supplementary Movie 1, when dual-focal beams are used, two bacterial clusters simultaneously start to move from the diagonal vertices of the rectangle, then along the boundary of the rectangle toward the other diagonal vertex of each. This simultaneous manipulation of multiple bacterial clusters fully demonstrates the ability of array-based acoustic tweezers to control the acoustic fields accurately in time and space. Moreover, GVs@E. coli can be used to display the transformation of the entire acoustic field patterns, based on their characteristic of clustering in areas of high acoustic intensity. A focusing vortex with various topological charges can be generated by the array using the holographic acoustic element framework method (HAEFM)[49]. In this method, the IB algorithm is applied to achieve the focusing element, and a spiral-patterned holographic signature with a changeable topological charge is used to create the vortex. A peculiar feature of vortex beams is the emergence of a pressure intensity minimum at the focal point, which is surrounded by a bright ring of high intensity[49]. When the preset topological charge $m$ changes from 1 to 3, the patterns of the vortex fields displayed by the bacterial cluster change accordingly (Fig. 3d, Supplementary Fig. 6a–c and Supplementary Movie 1). The distances between the strongest intensity peaks in the vortex fields with $m = 1, 2$, and 3 are approximately 0.9, 1.4 and 1.8 mm, respectively. Because of the angular momentum, the cluster can rotate continuously around the centre of the vortex, and the rotation velocity decreases with increasing topological charge.

### In vivo acoustic trapping of GVs@E. coli

To examine the in vivo bacterial trapping capability, we intravenously injected the fluorescence-stained control bacteria or GVs@E. coli into the tail veins of mice. A mouse skin-fold dorsal window chamber model was established in this study[37] and enabled the trapping process within blood vessels to be visualised directly in real time using microscopy (Fig. 4a). The results indicated that the GVs@E. coli could be trapped in the focal zone when the ultrasound was turned on, whereas the control bacteria failed to be trapped even after 180 s ultrasound action (Fig. 4b, c and Supplementary Movie 2). We also tried to trap the GVs@E. coli clusters in the blood vessels with different diameters from 110 to 200 μm. The results showed that the acoustic tweezer system could trap the GVs@E. coli but not the control bacteria in the vessels with various diameters within 20 s (Fig. 4d and Supplementary Movie 2). The GVs@E. coli cluster formation time decreased (Fig. 4e), and cluster size increased (Fig. 4f) with increasing blood vessel diameter, mainly because larger blood vessels can supply circulating bacteria more efficiently. Thus, our results clearly show that the array-based acoustic tweezers can effectively trap the genetically engineered bacteria in vivo.

### In vivo programmable acoustic manipulation of GVs@E. coli

Given the successful trapping of GVs@E. coli in the vessels in vivo, we further examined the feasibility of programmable manipulation of bacteria by the electronically steered acoustic beams. As demonstrated by Fig. 5a and Supplementary Movie 2, upon systemic administration of the GVs@E. coli, ultrasound enabled bacterial trapping to be turned on at the beam focus. Obvious bacterial aggregation could be observed in the focal area within the blood vessel (diameter:

210 μm) after 67 s of ultrasound exposure. Next, we drove the bacterial cluster to move back and forth along the blood flow and kept the bacterial cluster still for 30 s at specific node positions. The bacterial cluster could be driven at 10 μm s$^{-1}$ in the blood vessel, similar to the speed under in vitro conditions, and the repeated movement/stop cycle could be realised by controlling the electronically steered acoustic beams. Once the ultrasound was off, the bacterial clusters in the blood vessels were no longer manipulated by the ARF and were quickly dispersed by the blood flow. Thus, our results showed that the movement of bacterial clusters could be manipulated in strict accordance with the programme setting, with the satisfactory precision of the acoustic tweezers in both time and space. To test the feasibility of directed manipulation of the GVs@E. coli in the more complicated blood flow scene, we focused the acoustic beams on the fork vessels with blood flows in different directions and at different velocities. The path of focal spot motion is programmed by the acoustic tweezers according to the shape of the bifurcate vessels. As displayed in Fig. 5b and Supplementary Movie 2, the GVs@E. coli could gather into a cluster at the intersection of the focal spot and the blood vessel, and the bacterial cluster could selectively and sequentially pass through the bifurcation vessel according to the movement of the acoustic beam. Notably, the bacterial cluster was more obvious when moving against the blood flow than when moving along the flow, mainly because the GVs@E. coli were more likely to be washed away due to the superposition of velocity along the flow. These results further proved that the acoustic tweezer system still has a well-programmable manipulation ability in the in vivo complex flow environment. Moreover, the simultaneous manipulation of multiple clusters was demonstrated in a blood vessel (Fig. 5c and Supplementary Movie 2). Two bacterial clusters were trapped simultaneously in the same blood vessel and manipulated to move closer together or farther away from each other. When the two bacterial clusters were far apart, they could be clearly distinguished. The downstream bacterial cluster was not as clear as the upper cluster because the clusters were intercepted by the upstream focal beam. In addition, when the two focal beams were close to each other, the GVs@E. coli accumulated in the middle area of the cluster due to the interference of the waves.

### Acoustic aggregation of GVs@E. coli in the tumour

To explore whether acoustic manipulation can help the genetically engineered bacteria aggregate and resettle in a tumour, we established a subcutaneously transplanted 4T1 tumour model and intravenously injected the DiR-labelled GVs@E. coli after 7 days, followed by treatment with or without manipulation of genetically engineered bacteria at the tumour by acoustic tweezers. The In Vivo Imaging System (IVIS) showed that the tumour that received acoustic manipulation had higher fluorescence than that of the tumour which was not treated with acoustic manipulation, revealing that acoustic manipulation could help these bacteria aggregate and resettle into the tumour (Fig. 6a). Bacterial clone formation experiments involving plating the tumour homogenate further revealed that significantly more genetically engineered bacteria grew on the LB agar plate with 100 μg mL$^{-1}$ kanamycin, 2.54- or 4.24-times more than the amounts the tumours received with the genetically engineered bacteria but without acoustic manipulation at 1:100 or 1:1000 dilution, respectively (Fig. 6b, c). Histological staining of tumour sections further confirmed that more fluorescence-labelled bacteria (red) were present in the acoustically irradiated tumours than in the non-irradiated tumours (Fig. 6d). Collectively, these results demonstrate that the ultrasound could trap these genetically engineered bacteria and promote their enrichment in tumours.

Although genetically engineered bacteria are promising as gene or drug delivery carriers for biomedical applications, their biosafety remains a concern when promoting them into clinical translation. Therefore, we further evaluated their potential side effects after

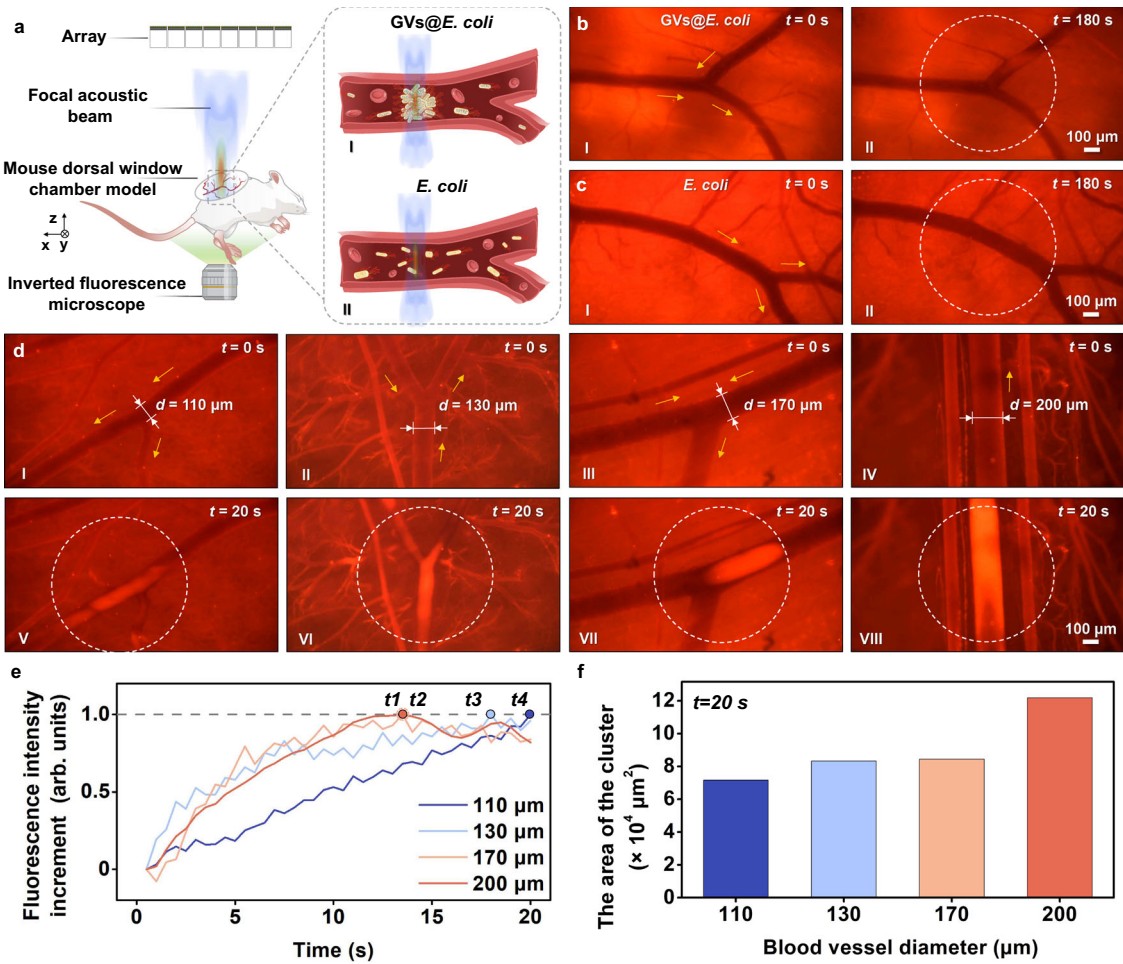

**Fig. 4 | In vivo acoustic trapping of GVs@*E. coli*. a** Schematic diagram of in vivo experiment setup. I and II are the schematic diagrams of the acoustic trapping processes of the GVs@*E. coli* and control *E. coli*, respectively. Comparison of acoustic trapping of **b** GVs@*E. coli* and **c** control *E. coli* in superficial blood vessels on the backs of mice. I in (**b**, **c**) indicates the microscopic images of blood vessels in the absence of ultrasound after injection of GVs@*E. coli* and control *E. coli*, respectively. II in (**b**, **c**) shows the microscopic images of blood vessels after exposure to ultrasound for 180 s based on the situation I in (**b**, **c**), respectively. Only GVs@*E. coli* can be trapped at the focal beam centre and form clusters in the vessels. **d** Acoustic trapping of GVs@*E. coli* in blood vessels of different diameters. I, II, III and IV are microscopic images of 110-, 130-, 170- and 200-μm-diameter vessels injected with GVs@*E. coli* before the ultrasound is on, respectively. V–VIII are microscopic images of trapped GVs@*E. coli* clusters in the corresponding blood vessels in cases I–IV after the ultrasound is turned on for 20 s. The yellow arrows, white dotted circles, symbol *d*, and *t* in (**b**–**d**) indicate the blood flow direction, focal zones, vessel diameter, and time, respectively. **e** The curves of normalised fluorescence intensity increment over time within the focal zones under vessels of different diameters in (**d**). *t1* to *t4* represent the moment when the maximum increment of fluorescence intensity is reached in 200-, 170-, 130-, and 110-μm diameter vessels, respectively. **f** The area of the GVs@*E. coli* cluster in vessels of different diameters (after the ultrasound is turned on for 20 s). The in vivo acoustic trapping of GVs@*E. coli* is shown in Supplementary Movie 2.

systemic administration of the GVs@*E. coli*, followed by manipulation with acoustic tweezers. As shown in Fig. 6e, f, haematological examination revealed that the peripheral blood (including leucocytes, lymphocytes, intermediate cells and granulocytes) and liver functions (including ALT, AST and GGT) did not experience significant changes in the GVs@*E. coli* + US group in comparison with the GVs@*E. coli* and PBS control groups. In addition, histological analysis with H&E staining demonstrated that systemic administration of the GVs@*E. coli* combined with acoustic manipulation did not cause any apparent pathological damage to major organs, including the heart, liver, spleen, lungs, or kidneys, in comparison with bacterial administration and PBS control groups (Fig. 6g). Collectively, all these data suggest that the acoustic tweezer system combined with the GVs@*E. coli* had good biosafety when applied in the in vivo condition.

### Acoustic aggregation of GVs@YB1 for tumour therapy

Next, we further evaluated the anti-tumour efficacy of acoustic manipulation of GVs@YB1 in vivo. The GV expression in YB1 (Supplementary Fig. 8a, b) and the acoustic manipulation of GVs@YB1

(Supplementary Fig. 9a–c) were first demonstrated. The acoustic trapping of GVs@YB1 and GVs@*E. coli* showed similar trends and it seemed to take longer to trap GVs@YB1 mainly because YB1 had flagellar motion. The treatment strategy was shown in Fig. 7a. The tumour-bearing mice were randomly divided into five groups, including PBS (Control), ultrasound alone (US), ultrasound combined with *E. coli* BL21 without GVs (*E. coli* + US), YB1 with GVs (GVs@YB1), and GVs@YB1 combined with ultrasound (GVs@YB1 + US). As shown in Fig. 7b, e, no significant tumour inhibitory effect was found in the US group, reaching more than 1500 mm³ mean tumour volume at day 17, similar with the Control group. Slight tumour inhibitory effects were observed in the *E. coli* + US and GVs@YB1 groups, with 1350 mm³ and 800 mm³ mean tumour volume at day 21, respectively. By contrast, the GVs@YB1 + US group exhibited the strongest tumour inhibitory effect, with only 360 mm³ mean tumour volume at day 21. No significant body weight changes were observed in these tumour-bearing mice during the treatment period (Fig. 7c). Notably, the GVs@YB1 + US group exhibited the longest survival time, achieving more than 45 days in comparison with other groups (Fig. 7d). These data indicated that the

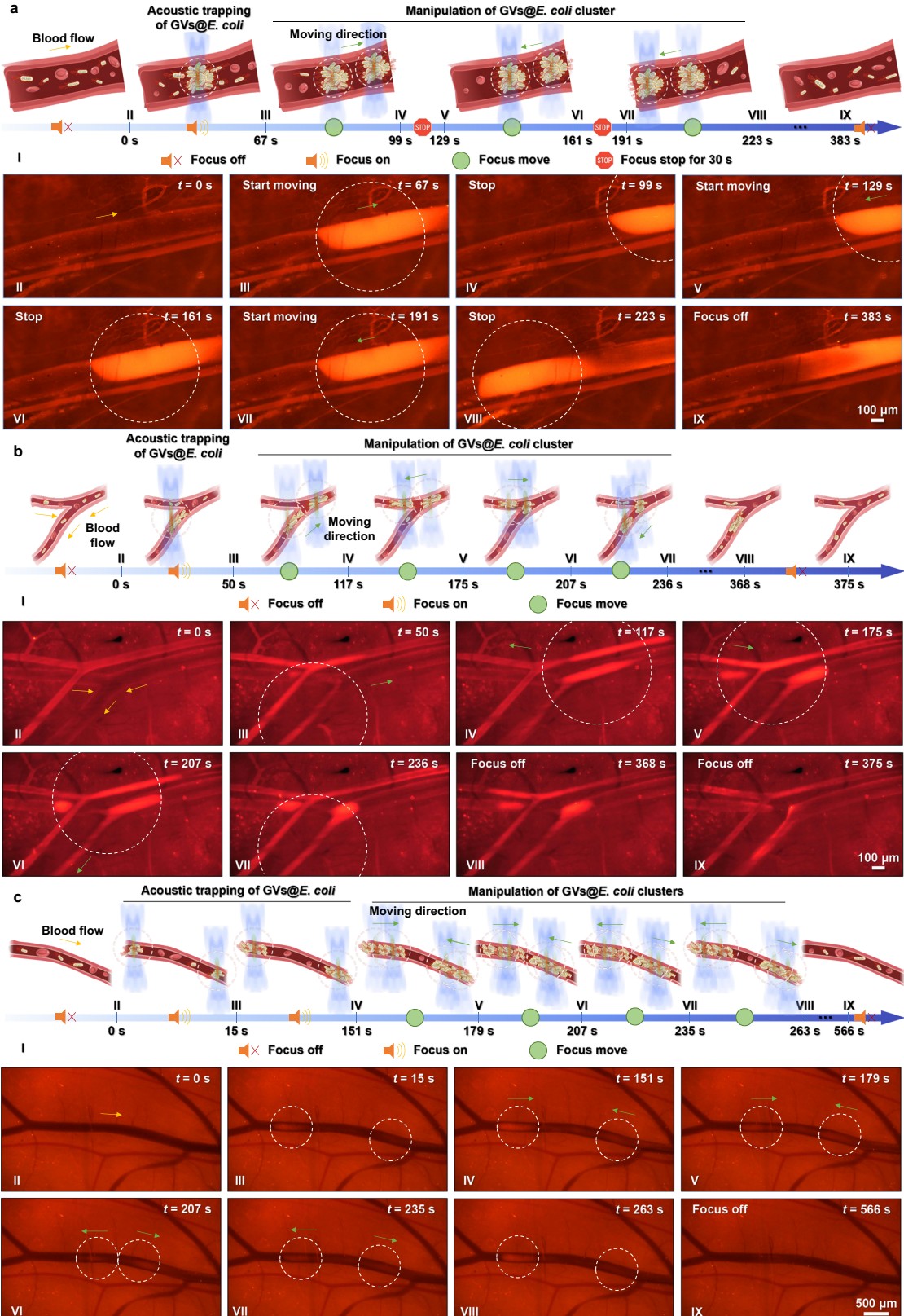

**Fig. 5 | In vivo programmable acoustic manipulation of GVs@E. coli.**
**a** Manipulation of the GVs@E. coli cluster to move back and forth along the blood vessel, while making the cluster stop moving for 30 s at specific node positions.
**b** Manipulating the GVs@E. coli cluster selectively through the fork of the vessels.
**c** Simultaneously manipulating double clusters in the same blood vessel and driving the clusters to move closer or farther away from each other. I in (**a**–**c**): Schematics of manipulation experiment setups. II–IX in (**a**–**c**): Time-lapse microscopic

fluorescence image sequences of the locations of the manipulated GVs@E. coli clusters in each case. The yellow arrows, green arrows, white dotted circles, symbol d, and t in (**a**–**c**) indicate the blood flow direction, translation directions of the clusters, focal zones, vessel diameter, and time, respectively. The in vivo programmable acoustic manipulation of GVs@E. coli is shown in Supplementary Movie 2.

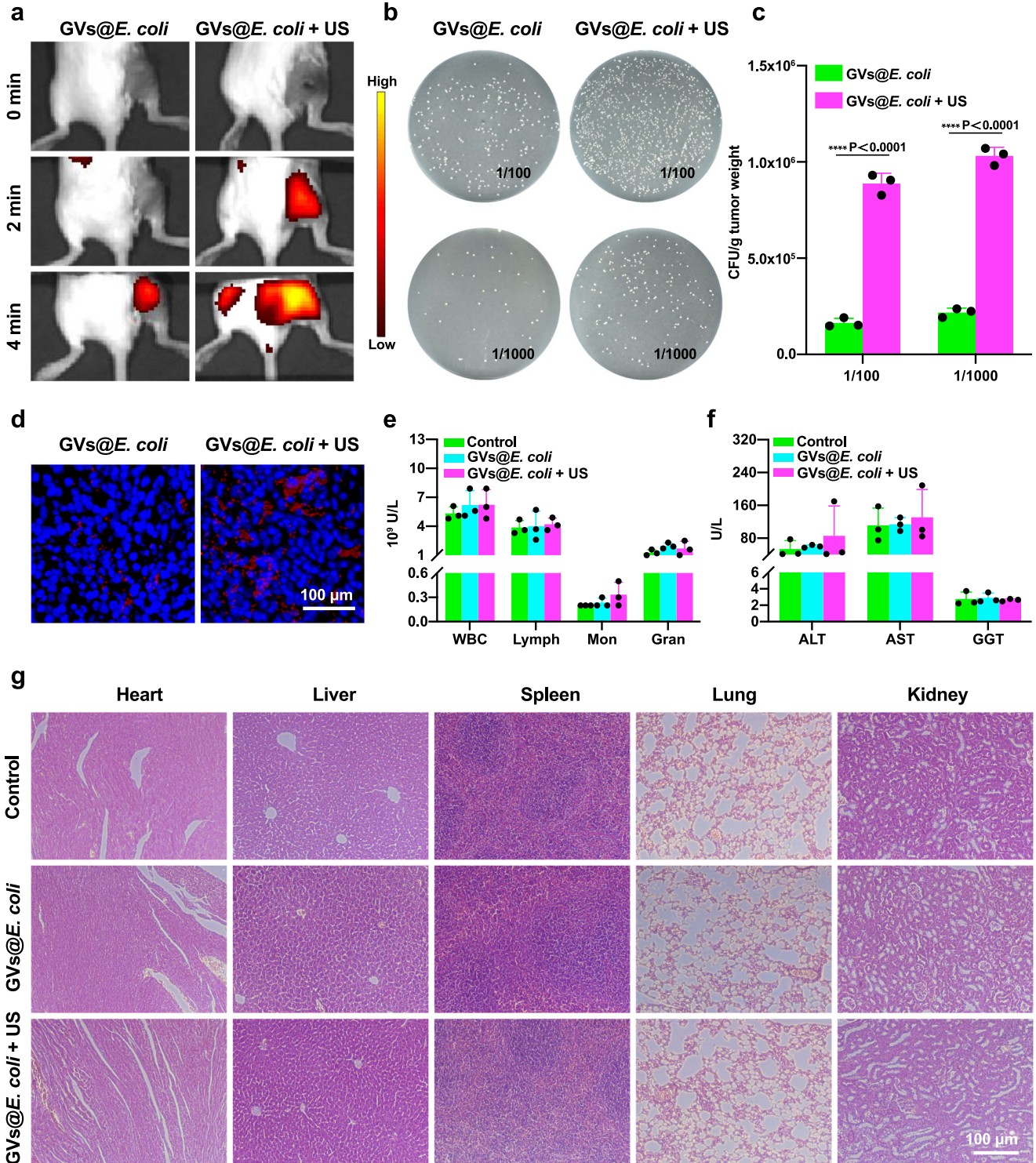

**Fig. 6 | Acoustic aggregation of GVs@*E. coli* in a tumour. a** Fluorescence imaging of tumour-bearing mice after injection of GVs@*E. coli* with or without acoustic manipulation for different durations. **b** Selective growth of the genetically engineered bacteria on the LB plates with 100 μg mL⁻¹ kanamycin at 1:100 or 1:1000 dilution of tumour homogenate. **c** Quantitative analysis of the bacterial counts in the tumour received with or without acoustic manipulation, *n* = 3 biologically independent samples per group. **d** Fluorescence microscopy images of tumour sections. The cell nuclei were stained with DAPI (blue), and the bacteria were labelled with DiR (red). **e** Haematological examination and **f** liver function test of mice after different treatments, *n* = 3 biologically independent samples per group. **g** H&E staining of the major organs (heart, liver, spleen, lung, and kidney). Images were representative of three experiments in (**d**, **g**). Data in (**c**, **e**, **f**) are presented as mean ± s.d. Statistical analysis was implemented by using the two-way analysis of variance with Sidak's test. **** means *P* < 0.0001. Source data are provided as a Source Data file.

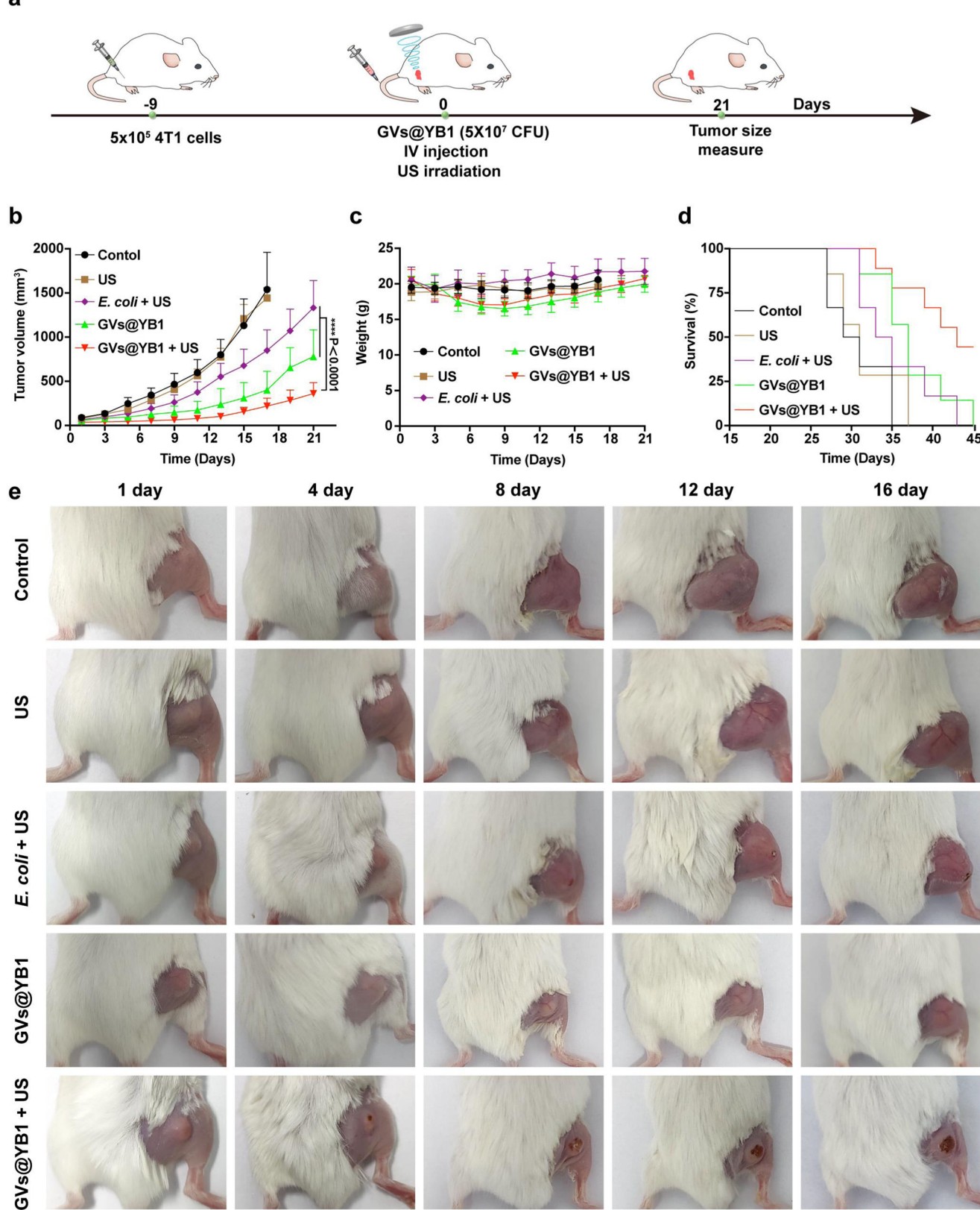

**Fig. 7 | Acoustic aggregation of GVs@YB1 improves the anti-tumour efficacy.**
**a** Schematic illustration of acoustic aggregation of GVs@YB1 to inhibit tumour growth in tumour-bearing mice. **b** Tumour growth curves of tumour-bearing mice in different treatment groups. **c** Body weight change curve of tumour-bearing mice in different treatment groups. **d** Survival curves for different treatment groups.

**e** Representative photos of tumour growth in each group of tumour-bearing mice after different treatments. Data in (**b**, **c**) are presented as mean ± s.d., and $n = 5$ biologically independent samples per group. Statistical analysis was implemented by using the two-way analysis of variance with Tukey's test. **** means $P < 0.0001$. Source data are provided as a Source Data file.

GVs@YB1 + US group could more effectively inhibit tumour growth and improve the survival time of tumour-bearing mice. Furthermore, we detected the GVs@YB1 aggregation in the tumour of tumour-bearing mice on day 1, day 7 and day 14 after treatment. Our results showed that the number of GVs@YB1 in the tumour of GVs@YB1 + US group was significantly higher than that in the GVs@YB1 group (Supplementary Figs. 10a, b and 11a, b).

## Discussion

In vivo cell manipulation will facilitate the development of targeted drug delivery, microbial diagnostics, cell-based therapy, etc. Common propulsion mechanisms such as light, magnetic, and acoustic manipulation strategies have been applied to drive cells externally. Unlike other propulsion mechanisms, acoustic tweezers can realise the non-contact manipulation of targets by utilising the momentum interaction between the acoustic waves and objects, presenting a perfect candidate for in vivo cell manipulation due to their advantages of high tissue penetrability, strong ARF, and feasibility in non-transparent media. However, normal cells are difficult to be manipulated directly because of their small sizes and the similarity between their acoustic impedances and those of their containing media. In this study, we produced numerous GVs in bacteria by using genetic engineering technology. The presence of GVs significantly reduced bacterial average density and increased compressibility, thus greatly enhancing the sensitivity of the whole bacterium to ultrasound and tens of times increasing ARF exerted on the bacterium. Moreover, the in-phase oscillation of GVs of adjacent bacteria significantly increased secondary radiation force, which also facilitated GVs@*E. coli* to aggregate. As a result, these genetically engineered bacteria can overcome Stokes force and be trapped or driven by the ARF. More importantly, the genetically engineered bacteria can be manipulated by acoustic tweezers in a programmable manner by virtue of electronically steered acoustic beams or mechanical movement of the array transducer (or microscope stage, relatively) built in the acoustic tweezers (Supplementary Fig. 7a, b and Supplementary Movie 2).

Indeed, the emergence of genetic engineering and biosynthetic technologies has opened up many possibilities for cell modification and can be adapted to various application scenarios by combination with ultrasound. For example, genetically encoded GVs can also be used in ultrafast amplitude-modulated molecular and hemodynamic ultrasound imaging[50], digital holographic microscopy based on phase contrast[51], non-invasive biological imaging[52], and other applications. In addition to generating GV structures, bacteria or CAR-T cells can be genetically engineered into temperature-driven gene expression circuits, through which these genetically engineered cells or bacteria can selectively express specific therapeutic genes in the diseased local tissue via the heating effects of focused ultrasound, thus avoiding systemic side effects or immune responses to bodies[53]. Thus it can be seen that genetic engineering and biosynthetic technologies provide very powerful means of cell modification, and their combination with ultrasound will open up many new research directions for biomedical applications.

In this study, we chose *E. coli* as the object of the acoustic manipulation, mainly based on the following facts: (1) *E. coli* BL21 is an all-purpose strain used for high-level protein expression for genetic engineering, and successful biosynthesis of GVs has been achieved in this strain[43]; (2) *E. coli* BL21 is relatively safe when systemic administration is performed. In fact, numerous scholars have utilised genetically engineered *E. coli* BL21 for tumour treatment[54,55]. As excellent gene delivery carriers, some bacteria were specially modified to enhance their tumour-targeting capabilities or were genetically engineered to improve their survival in the tumour immunosuppressive environment[56]. Tumour suppressor proteins could also be expressed in these genetically engineered bacteria to alleviate their potential side effects or enhance their anti-tumour efficacies. In this study, we used

acoustic tweezers to manipulate the genetically engineered bacteria and realised trapping and programmable movement in vitro and in vivo. Directed manipulation of the movement of *E. coli* in the circulatory system, such as in the blood and lymph nodes, has significant research and application value. For example, in cancer treatment, the abundant accumulation of genetically engineered bacteria in tumour neovascularization may lead to thrombosis, inhibiting tumour growth. In addition, a local increase in the concentration of the bacteria in the tumour area may activate the autoimmunity of the basal body and attract immune cells to accumulate at the tumour site. This study already proved the feasibility of increasing the aggregation efficiency of the genetically engineered bacteria in tumour areas by using acoustic trapping, which may be beneficial for tumour treatment. The acoustic manipulation of genetically engineered bacteria under in vivo conditions has been demonstrated by our study. And considering the advantages of ultrasound in terms of high tissue penetrability and focusing ability, remote in vivo acoustic manipulation technology shows promise for application to other therapeutic cells such as CAR-T cells, natural killer cells, or stem cells in future clinical applications.

## Methods

### Preparation of GVs@*E. coli* and GVs@YB1

The study complied with all relevant ethical regulations, which were approved by the Institutional Animal Care and Use Committee at Shenzhen Institute of Advanced Technology (approval number: SIAT-IACUC-210303-YGS-YY-A1700). The competent *E. coli* BL21 (A1) bacteria (AngYuBio, Catalog No. G6024) were transformed using the pET28a_T7-ARG1 plasmid (Addgene, Catalog No. 106473) by applying the chemical transformation protocol. A monoclonal bacterium was picked up and seeded in LB liquid medium containing 100 μg mL$^{-1}$ of kanamycin (Aladdin, Catalog No. K103026), 1% glucose (Aladdin, Catalog No. G116303) for 16 h at 37 °C. Then, the bacteria were further expanded culture in LB liquid medium with 100 μg mL$^{-1}$ of kanamycin, 0.2% glucose at 37 °C until the concentration of bacteria reached OD$_{600nm}$ = 0.4−0.5. After that, 0.5% L-arabinose (Sigma, Catalog No. V900920) and 0.4 mM IPTG (YEASEN, Catalog No. 10902ES08) were added to the bacteria to induce the expression of GVs for 22 h at 30 °C. The genetically engineered bacteria were collected in a 50-mL tube by centrifugation 350 × *g* at 4 °C for 3 h. For GVs@YB1, the pDT103 (Addgene, Catalog No. 106475) was transformed into *salmonella Typhimurium* YB1 (YB1, gift from Chenli, Liu) by using the same protocol. The culture process of transformed YB1 was similar with *E. coli* BL21, except the diaminopimelic acid (DAP, Sigma, Catalog No. D1377) with a final concentration of 50 μg mL$^{-1}$ was added to the LB liquid medium, and 3 nM N-(β-ketocaproyl)-l-homoserine lactone (AHL, Sigma, Catalog No. K0037) was added when the concentration of YB1 culture achieved 0.3 OD$_{600nm}$ to induce the expression of GVs.

### Characterisation of GVs@*E. coli*

The genetically engineered bacteria before and after being induced by L-arabinose and IPTG for 22 h were visualised by phase-contrast microscopy (Olympus IX83, Olympus Corporation, Japan) at ×1000 magnification to confirm GVs production. Moreover, these bacterial samples were examined by adding them to the copper mesh, using 3% phosphotungstic acid (pH = 7.0) to dye these samples. After air drying, the bacteria were observed via TEM (80 V, Hitachi H-7650).

### Development of 2D planar ultrasound array

A 3 MHz, 64-element (8 × 8) planar array was designed and fabricated for acoustic manipulation of GVs@*E. coli*. The whole array aperture was 6 mm, and the centre-to-centre spacing between adjacent elements was 0.75 mm. The specific manufacturing steps were as follows. By using a dice-and-fill technique, the piezoelectric 1−3 composite was generated from a piece of PZT-5H bulk ceramic. The composite was lapped to the desired thickness, which was determined using a

PiezoCAD simulation. A chrome/gold electrode (thickness: 200 nm/500 nm) was sputtered on both polished surfaces of the composite. An acoustic matching layer (component: aluminium oxide powder (23 μm) and Epo-Tek 301 epoxy (Epoxy Technologies, MA, USA)) was cast on the front surface of the composite. The matching layer, which was cured at room temperature for 24 h, was polished to an optimal thickness of 250 μm. The prepared acoustic stack was diced with the matching layer, and the designed 64 elements with total dimensions of 6 mm × 6 mm could be obtained. Then, elements with dimensions of 0.75 mm × 0.75 mm were diced on the back surface of the composite. A single customised 64-electrode polyimide flexible circuit (trace width: 254 μm, thickness: 118 μm) was pasted onto the elements using Epo-Tek 301 epoxy. An acrylic housing was employed to fix the elements, which were attached to the flexible circuit, and the gap between the acoustic stack and housing was filled by applying an insulating epoxy. In total, 64 coaxial cables were connected to the flexible circuits. Finally, the housing was filled with a backing layer (component: Epo-Tek 301 epoxy with aluminium nitride powder (3–5 μm)).

### Formation and characterisation of the acoustic field

The IB algorithm and HAEFM were used to calculate the amplitude and phase information of each element in multi-foci and vortex acoustic fields, respectively, and the corresponding fields could be obtained by exciting the array with the calculated signals. A Verasonics Vantage 256 System (Verasonics Inc., WA, USA) was selected as the excitation system.

Immersed in water, the array was driven at a working frequency of 3 MHz with a corresponding wavelength (λ) of 500 μm. The focal acoustic field (focal depth: 6 mm, f-number = 1) was simulated by using COMSOL Multiphysics, which is a commercial finite-element software, and scanned by a needle hydrophone (diameter: 0.2 mm) attached to a 3D positioning system (Precision Acoustics Ltd., Dorchester, UK).

### ARF calculation

For a microsized spherical particle whose diameter is much smaller than the wavelength, the gradient of the Gorkov potential ($U$)[40,49] can be used to calculate the ARF of the particle:

$$\mathbf{F} = -\nabla U, \tag{1}$$

where $U$ can be expressed by the complex acoustic pressure (**p**) and its spatial derivatives:

$$U = a_1(|\mathbf{p}|^2) - a_2(|\mathbf{p_x}|^2 + |\mathbf{p_y}|^2 + |\mathbf{p_z}|^2), \tag{2}$$

$$a_1 = \frac{1}{4}V\left(\frac{1}{c_0^2\rho_0} - \frac{1}{c_p^2\rho_p}\right), \tag{3}$$

$$a_2 = \frac{3}{4}V\left(\frac{\rho_0 - \rho_p}{\omega^2\rho_0(\rho_0 + 2\rho_p)}\right). \tag{4}$$

Here, $\rho$ is the density, $c$ is the sound velocity (the subscripts $0$ and $p$ refer to the medium and particle material, respectively), $\omega$ is the angular frequency of the incident wave, and $V$ is the particle volume. The length of the bacteria is assumed to be 3 μm. The density and sound velocity of the GVs are 120 kg m$^{-3}$ (see ref. 57) and 733 m s$^{-1}$ (see ref. 58), respectively, and the density and sound velocity of $E.\ coli$ are 1079 kg m$^{-3}$ (see ref. 59) and 1531 m s$^{-1}$ (see ref. 45), respectively. By assuming that GVs occupy 10% of the intracellular space (typically, GVs occupy 8–16% of the volume of a single bacterium), the volume-averaged density and compressibility ($\beta = \frac{1}{\rho c^2}$)[60] can be calculated. Thus, the density and sound velocity of the GVs@$E.\ coli$ can be obtained as 983 kg m$^{-3}$ and 731 m s$^{-1}$, respectively.

### System integration for in vitro experiments

Firstly, $200 \times 10^7$ CFU mL$^{-1}$ bacteria with or without GVs were obtained by low-speed centrifugation with $350 \times g$ at 4 °C for 3 h and then stained with 1, 1′-dioctadecyl -3, 3, 3′, 3′-tetramethylindocarbocyanine perchlorate (DiI, Beyotime Biotechnology, Catalog No. C1991S) for 20 min in the dark to obtain the DiI-labelled $E.\ coli$ and GVs@$E.\ coli$. The trapping dynamics of DiI-labelled $E.\ coli$ or GVs@$E.\ coli$ by acoustic tweezers were observed and recorded from the bottom view using an inverted microscope (IX73, Olympus Corporation, Shinjuku, Tokyo). For the bottom-view observations, the array was aligned with the objective in a face-to-face arrangement. A 500-μm inner diameter silicone tube, a T-shaped PDMS cavity with 500 μm width, and an ordinary 10-cm-diameter Petri dish were respectively positioned at the focus between the array and microscope for the in vitro experiments under different conditions. The total size of the trapped bacterial cluster for each set of parameters was estimated from the acquired images or videos using ImageJ software.

### In vivo manipulation experiment setup

The female BALB/c mice (weight of 18–20 g and aged 6–8 weeks) were provided by Guangdong Yaokang Biotechnology Co., Ltd. Before the experiments, the mice were anaesthetised with isoflurane (2.5% for induction and 1.5% for maintenance) using a gas anaesthesia ventilator (R530, RWD Life Science Co., Ltd., Shenzhen, China). Two titanium plates and a cover-glass window (80340-1630, Citotest Scientific Co., Ltd., Nanjing, China) were mounted on the mouse dorsal skin and were constructed as the window chamber model. The DiI-labelled $E.\ coli$ and GVs@$E.\ coli$ (concentration: $200 \times 10^7$ CFU mL$^{-1}$, dose: 125 μL each time) were systemically administrated by tail vein injection. The array transducer was positioned on the window chamber model with the gap is filled with an ultrasound coupling agent. During the in vivo experiments, the input voltage was increased to 20 V (duty cycle: 10%), so that sufficient ARF could be maintained through the tissue of the mouse.

### Establishment of tumour model and tumour treatment

Female BALB/c mice aged 4-6 weeks were selected to establish a breast tumour model. Briefly, mice were kept in a Specific-pathogen-free (SPF) environment at 23–25 °C on a 12-h light/dark cycle and provided with adequate water and food. 100 μL phosphate buffered saline solution (PBS, Gibco, Catalog No. 10010023) containing $5 \times 10^5$ 4T1 breast cancer cells (Procell, Catalog No. CL-0007) was injected subcutaneously into the right thigh of mice. When the tumour volume reached 50 mm$^3$, these tumour-bearing mice were randomly divided into five groups, including control group (intravenous injection of 100 μL PBS), US group (ultrasound irradiation for 20 min), $E.coli$ + US group (intravenous injection of $5 \times 10^7$ CFU $E.coli$ BL21 without GVs and then ultrasound irradiation for 20 min), GVs@YB1 group (intravenous injection of $5 \times 10^7$ CFU GVs@YB1), GVs@YB1 + US group (intravenous injection of $5 \times 10^7$ CFU GVs@YB1 and then ultrasound irradiation for 20 min). Acoustic parameters were set at 20 V input voltage and 10% duty cycle. The tumour volume was measured every two days. When the tumour volume exceeded 2000 mm$^3$ or the experiment ended, these tumour-bearing mice were euthanized by carbon dioxide asphyxia.

### Acoustic aggregation of engineering bacteria in the tumour

Bio-distribution in vivo: For the fluorescence imaging in vivo, the DiR (1,1′-dioctadecyl-3,3,3′,3′-Tetramethylindotricarbocyanine Iodide, Invitrogen™, Catalog No. D12731) was used to label the bacterial membrane. Briefly, 60 μL DiR (1 μM) was added to the bacteria suspension (OD$_{600nm}$ = 0.5) for co-incubation at 37 °C for 60 min. Then, the bacterial mixture was washed with PBS twice to remove the free DiR. After that, 100 μL of the DiR-labelled GVs@$E.\ coli$ ($5 \times 10^7$ CFU) was intravenously injected into mice. The tumour was irradiated by ultrasound for

2 min or 4 min. Next, the tumour was imaged by IVIS after ultrasound irradiation. The irradiated and non-irradiated tumours were collected. Half of each tumour was cut into sections for histochemical staining, and the other half was weighed and homogenised in sterile PBS (pH = 7.2). These tumour homogenates were diluted (100 or 1000 times), plated on the LB plates with 100 µg mL$^{-1}$ kanamycin, and cultivated overnight. The bacterial colonies were counted after 12 h, which was counted by the ImageJ software, and the statistical condition of "size=1-intensity" was set in order to exclude satellite colonies.

**Biosafety assay**

Nine mice were used for biosafety assay, six mice were intravenously injected with the GVs@*E.coli* at the dose of $5 \times 10^7$ CFU per mouse and three of them were irradiated by ultrasound as described above. Three healthy mice without any treatment were used for the control group. Blood samples were collected at 7 days for the haematological examination and liver function test. Furthermore, the major organs of these mice, including the heart, liver, spleen, lungs and kidneys, were collected for H&E staining.

**Statistical analysis**

Herein, the GraphPad Prism software was used for all statistical evaluations. Each mean and standard error value presented in the figures are calculated based on at least three independent measurements. The results were significant when ****$P < 0.0001$; ***$P < 0.001$; **$P < 0.01$; *$P < 0.05$.

**Reporting summary**

Further information on research design is available in the Nature Portfolio Reporting Summary linked to this article.

## Data availability

All relevant data supporting the key findings of this study are available within the article and its Supplementary Information files or from the corresponding author upon reasonable request. Source data are provided with this paper.

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

## Acknowledgements

The authors acknowledge the support of Mr. Shuo Wang, Mr. Zeping Gao, Mr. Xin Xu, Ms. Ying Zhou, Ms. Bing Wang and Ms. Siyi Lan from Shenzhen Institute of Advanced Technology, for the cell and therapeutic experiment. This work was supported in part by the National Key R&D Program of China (2020YFA0908800 and 2020YFA0211400), National Natural Science Foundation Grant of China (81527901, 61571431, 11774370, 81827807, 81827802 and 61971411), Shenzhen Peacock Plan (KQTD20170810160424889), Scientific Instruments Funding of Chinese Academy of Sciences (YJKYYQ20190077), Shenzhen Outstanding Scientific and Technological Innovation Talents (Distinguished Young Scientists, RCJC20210609104443085), Shenzhen Key Technology Project (JSGG20210713091811037, JSGGZD20220822095200002), Advanced Medical Imaging Key Technology and Core Component Innovation Team (JSGGKQTD20210831174329010), the China Postdoctoral Science Foundation (2022M723305), and Shenzhen Key Laboratory of ultrasound imaging and therapy (ZDSYS20180206180631473).

## Author contributions

T.M., F.Y. and H.Z. conceived the idea and designed the research methodology. T.M., J.H., Y.L. and Q.Z. designed and fabricated the device. Ye Y. and Yaozhang Y. performed the experiment and analyzed the data with the help of D.L., M.L. and Y.W. under the supervision of T.M., F.Y. and H.Z. Ye Y. and Yaozhang Y. wrote the manuscript, and T.M., F.Y. and H.Z. contributed to the revision.

## Competing interests

The authors declare no competing interests.
