## [Peer Review File · Nature Communications]

REVIEWER COMMENTS

Reviewer #1 (Remarks to the Author):

This paper describes the enhancement of acoustic tweezer using a specific bacterium (GVs@E. coli) that can release small microbubbles. Clusters of very small GVs (c. 3 μ m) are caused to agglomerate in much larger assemblies (c. 100s of μ m). These assemblies are first manipulated in PDMS channels and then within the blood vessels in a live mouse. It was then shown that GVs could be delivered to a cancerous tumour in a mouse without damage to major organs. The approach works because the GVs release and trap microbubbles that act to significantly increase the acoustic contrast. The authors show that this contrast increase is sufficient to enable the agglomeration and selective movement of GV-clusters, whereas similar unmodified bacteria are not trapped due to their low contrast and small size. The GV concept already existed and was developed for ultrasound imaging contrast enhancement. In Ref [41] its use in acoustic manipulation has been demonstrated. The current paper extends this by a) implementing the manipulation using a phased array which allows movement of the trap without physical movement of the transducer, and b), most significantly, implementing the approach in vivo. The in vivo scenario explored is in blood vessels with normal blood flow and so is a very important step forwards as this is a challenging environment with competing forces due to the flow. The fact that they show successful manipulation in the blood vessels opens the door for other in-vivo applications such as acoustically manipulated targeted drug delivery. Hence, the paper describes and import step forwards in acoustic tweezing and targeted drug delivery and will be widely read and build upon in future years.

The authors should address the follow relatively minor issues.

- 1) P5, line 85, where the novel step of the GVs is described, it would be good to quantify the improvement in terms of contrast and force.
- 2) When the GVs are described, line 89 it would be good to mention their size.
- 3) P6 it would be good to comment that the acoustic field shape was simulated (my understanding, please correct if wrong).
- 4) P7, line 129 (and line 286) it is hypothesised that the main competing force for the bacteria is Brownian motion. I'd have thought it was flow induced drag? Some evidence of the Brownian motion hypothesis is needed.
- 5) Around this same region it would be useful to explain that the mechanism of trapping is one of changing the local average mass density. This would make it clear that the action is a combined effect of many GVs, rather than individual GVs.
- 6) P10, line 190 onwards, describes the use of a vortex to rotate the clusters. Can the authors comment on the purpose of this aspect? Are there applications that could benefit from this, or are they just demonstrating an interesting capability?
- 7) P11, line 208, it would be good to estimate how many GVs are in a typical cluster.
- 8) P 19, line 383, how is it known that the microbubbles occupy 10% of the space? This seems to be quite speculative. Hence it might be better to give a range of plausible values and quote upper and lower limits.
- 9) In figure 4 and 5 it would be helpful to mark the focal point. Do the circles in fig 5 represent the focal zone? It would be good if they did and in any case, it would be helpful to have this information non fig 4 as well.

Reviewer #2 (Remarks to the Author):

The manuscript by Yang et al describes the in vivo manipulation of gas vesicle expressing E. Coli using a programmable acoustic array. Building on previous findings that GV expression in bacteria allows them to experience strong acoustic radiation force, this study shows that such bacteria can be

manipulated with an electronically programmable array, that such manipulation works within the vasculature in mice, and that this allows systemically injected bacteria to be acutely concentrated in subcutaneous tumors. This is an interesting study that merits consideration.

On the question of significance, reference 41 (Wu et al) already introduced the concept of GV-based acoustic manipulation and demonstrated that GV-expressing E. Coli can be selectively manipulated by ultrasound *in vitro*, including being trapped at an acoustic focus and moved around in solution. As far as I can tell, Yang et al also used the same bacterial genetic constructs that were used in that study. The authors misrepresent the contents of ref 41 by citing it as an "imaging" study in the introduction on line 89. They reference this study again on line 128 when discussing the acoustic contrast factor of GV-expressing E. Coli, but again in a way that obscures its key results. I think the authors should modify their introduction to more honestly cite and describe this study, from which they clearly drew inspiration for their own work.

Having said that, I don't view the existence of Wu et al as a total deal-breaker for the publication of this manuscript because this new work advances into acoustic manipulation of bacteria *in vivo**, which was not previously demonstrated. However, I think the *in vivo* demonstration of this technique and its utility needs to go considerably further for it to be suitable for a journal like Nat Comms. The basic demonstration that focused ultrasound can trap and move cells inside blood vessels is not a sufficiently large step on its own, in my view, because it is straightforwardly expected from *in vitro* results. On the other hand, the application of acoustic trapping to enhancing tumor treatment would be sufficiently more significant. However, I think this demonstration needs to be much more convincing.

As of now, the tumor demonstration only shows that acutely after cell administration, focused ultrasound can concentrate the bacteria in tumors. However, whether this very short-term effect would have any impact on anti-tumor probiotic therapy is uncertain. Tumor-homing bacteria have a well-established ability to colonize tumors on their own, and are thought to initially penetrate in small numbers and rapidly proliferate through cell division within the immunosuppressed regions of the tumor. While aggregating the bacteria at the tumor site at the time of administration could potentially give this process a boost, this is not necessarily the case. For example, bacteria concentrated in major vessels may not enter the correct niche inside the tumor, and aggregation could conceivably even be harmful to their ability to do so. Furthermore, having a large local bolus of bacteria could stimulate the immune system against the bacterial agent, reducing its long-term persistence. To demonstrate that the *in vivo* manipulation of bacterial agents is actually beneficial in tumor treatment, it would be necessary to measure bacterial colonization over several weeks with and without ultrasound and monitor its effects on tumor growth and animal survival. This is standard in other tumor-homing bacteria studies (e.g. see studies by Tal Danino or Jung-Joon Min). Furthermore, I think it is essential to include controls for ultrasound treatment combined with non-GV-expressing E. Coli because ultrasound by itself could have a damaging or immunogenic effect on the tumor (e.g. through heating).

Bacterial manipulation using a programmable array is also a new contribution relative to the previous use of single-element transducers and standing waves. However, since people have used such arrays to manipulate other particles, I don't think this by itself is a sufficient advance to compensate for insufficient *in vivo* data.

Some minor points:

How do the authors explain the concentration dependence of cell cluster size in Fig. 2n? Are the clusters large enough that they take up all the bacteria available, is it a kinetic effect (in which case it would be interesting to see what happens after longer exposure), or are there some inter-cell acoustic or chemical interactions?

Fig 4d(I) appears to show bacterial cells lining the vessel walls before the ultrasound is applied. Is this so, and do the authors have an explanation for why it should be the case?

Is there a way to make the results in Fig 4 and 5 more quantitative? What is the success rate of trapping attempts? How close or far from the predicted focus do the aggregates show up? What is the size of the aggregates axially and across the vessel as a function of the vessel diameter and ultrasound parameters?

The authors used BL21 E. coli, which lack flagella. Would acoustic trapping and manipulation work as well with motile strains, like the kind most commonly used in tumor-homing therapies?

One line 35, I think the authors mean "CAR-T" instead of "CAT-T".

Reviewer #3 (Remarks to the Author):

Comments to the authors

=====

In the manuscript, the authors demonstrated the in-vivo manipulation of genetically engineered bacteria using electronically steered acoustic beams developed by an acoustic transducer array. The results are remarkable and the study is data-rich by comparisons between engineered and nature's E. coli, as well as in-vitro and in-vivo conditions. The control trapping experiments with and without GVs inside bacteria show successful gas based acoustic navigation. The manuscript is well-organized and easy to understand. The authors showed controlled manipulation of the clusters using transducer array. Here I have some comments for you to consider:

1) "... living cells are difficult to acoustically manipulate because of their small size and the similarity between their acoustic impedance and that of the medium" "Second, the acoustic impedance of the cells is very close to that of the medium, resulting in less ARF generation". Citing references and quantifying the acoustic impedance of "normal cells" and "the medium" will be adequate. I suggest to cite this paper, which describes the effectiveness of microbubble manipulation in artificial vasculature (<https://doi.org/10.1002/admi.202200877>).

2) "The maximum peak-to-peak acoustic pressure in the focal zone was measured to be approximately 1.68 MPa." This level of acoustic pressure will not only destroy gas microbubbles but may induce inertial cavitation. Could the authors discuss the safety of bacteria with engineered GVs in the blood vessels? Additionally, in Figure 5a, the cluster size is significantly larger than the vasculature's diameter. Will it block the blood flow? Will it induce platelet aggregation or thrombosis?

3) What is the speed of GVs in the vasculature? Can the authors characterize the velocity plot of bacteria in mice vasculature? How long does the GVs stay within bacteria? Does it grow bigger over time? The latter is not relevant to the project; however, it will be nice to consider the rectified diffusion of microbubbles/GVs inside bacteria for further projects.

4) "However, further increasing the duty cycle from 7% to 10% did not significantly decrease the formation time (Fig. 2m)" why? The acoustic intensity is proportional to the input.

5) "The bacterial cluster is not generated at a concentration of 6.25×10^7 cells/ml" why lower concentration cannot result in aggregation? Also, is it a limitation the fact that you need higher bacterial concentrations to see clustering? What concentrations of bacteria would be toxic for the animal?

6) Can the authors explain the maximum cluster size? Also, will the bacteria cluster block the blood

flood? Can the authors give some insight?

7) Will these trapped E.coli escape from the focus point? What is the minimum trapping force require to trap and manipulate bacteria cluster?

8) "When the preset topological charge m changes from 1 to 3" what is changed? Parameters, phased configuration, or amplitude? What is the difference?

9) "the rotation velocity decreases with increasing topological charge" why?

10) "shows promise for application to other therapeutic cells such as CAR-T cells, natural killer cells, or stem cells in future clinical applications"? How are GVs generated in these cells?

11) What is the acoustic attenuation through air, skin, tissue, and vessel?

12) Does the transducer array touch the body? What is the thermal effect?

13) In fig. 2g and 2h, how the force direction is defined? Why a negative contrast factor resulted in the force directed toward the focus?

14) There is almost no references in the manuscript regarding acoustic manipulation of gas based vesicles. This is the main principle for the manipulation of bacteria, and it's not the first time that has been done. Please cite the following articles In vivo acoustic manipulation of microparticles in zebrafish embryos , Sci. Adv. 2022, 8, eabm2785.

Reviewers' comments:

Reviewer #1 (Remarks to the Author):

This paper describes the enhancement of acoustic tweezer using a specific bacterium (GVs@*E. Coli*) that can release small microbubbles. Clusters of very small GVs (c. 3 μ m) are caused to agglomerate in much larger assemblies (c. 100s of μ m). These assemblies are first manipulated in PDMS channels and then within the blood vessels in a live mouse. It was then shown that GVs could be delivered to a cancerous tumour in a mouse without damage to major organs. The approach works because the GVs release and trap microbubbles that act to significantly increase the acoustic contrast. The authors show that this contrast increase is sufficient to enable the agglomeration and selective movement of GV-clusters, whereas similar unmodified bacteria are not trapped due to their low contrast and small size. The GV concept already existed and was developed for ultrasound imaging contrast enhancement. In Ref [41] its use in acoustic manipulation has been demonstrated. The current paper extends this by a) implementing the manipulation using a phased array which allows movement of the trap without physical movement of the transducer, and b), most significantly, implementing the approach *in vivo*. The *in vivo* scenario explored is in blood vessels with normal blood flow and so is a very important step forwards as this is a challenging environment with competing forces due to the flow. The fact that they show successful manipulation in the blood vessels opens the door for other *in-vivo* applications such as acoustically manipulated targeted drug delivery. Hence, the paper

describes and import step forwards in acoustic tweezing and targeted drug delivery and will be widely read and build upon in future years.

Response:

Thanks for reviewer's insightful comments. Due to the small size (typically only a few microns) of cells and their close acoustic impedance to the medium (typical value is between $1.6-1.8 \times 10^6$ rayls vs 1.5×10^6 rayls for water) and complex *in-vivo* environment (the influence of normal blood flow and heterogeneous tissue), it is very difficult to manipulate cells acoustically in the *in-vivo* environment. The development of array-based holographic acoustic tweezers and the reported use of GVs to enhance the ultrasonic sensitivity and manipulation performance of cells have made it possible to realize the *in-vivo* acoustic manipulation of cells. As the reviewer said, our paper expands two aspects based on the GV's use in acoustic manipulation reported in *Reference 41*. First, we demonstrate the ability to manipulate GVs@*E. coli* using array-based holographic acoustic tweezers. These holographic acoustic tweezers have a very powerful ability to programmatically adjust the acoustic beams in both time and space, thus allowing movement of the acoustic traps without physical movement of the transducer and transformation of the acoustic fields. And by using the holographic acoustic tweezers, the GVs@*E. coli* were successfully trapped, translated, rotated and used for displaying the acoustic field transformation in this paper. Second, we further implemented the acoustic manipulation of the GVs@*E. coli in vivo*. Holographic acoustic tweezers based on single-side arrays have been proved to be one of the most suitable methods for acoustic manipulation *in vivo* due to their flexible

experimental settings and powerful manipulation capabilities. Herein, we have successfully demonstrated the acoustic manipulation of the genetically engineered bacteria (whose ultrasound sensitivity was enhanced by the GVs) in mouse blood vessels. This *in-vivo* acoustic manipulation of cells, which was realized in a challenging environment with normal blood flow, will greatly promote the development of acoustically manipulated targeted drug delivery, tissue repair and reconstruction and other *in-vivo* applications. In addition, we further demonstrated in the revised paper that trapping GVs@YBI at the tumour site using holographic acoustic tweezers can significantly improve the migration and resettlement of bacteria at the tumour site and effectively slow down the growth rate of the tumour, which will provide new ideas and methods for improving the efficacy of cellular immunotherapy for tumour treatment. The detailed responses are given in a point-by-point manner below and appropriate changes have been made in accordance with the reviewer's suggestions. In the revised manuscript, revisions have been highlighted in yellow. Thanks again to the reviewer for insightful suggestions and questions, which are of great help to the improvement of our manuscript.

The authors should address the follow relatively minor issues.

1) P5, line 85, where the novel step of the GVs is described, it would be good to quantify the improvement in terms of contrast and force.

Response:

Thanks for reviewer's constructive comments. We have added some quantitative data on typical acoustic impedance values for cells and medium, the improvement in terms of acoustic contrast (increased by 15 times) and the maximum simulated ARF (increased by 63 times). The acoustic impedance values are from reference [1] and [2], which the corresponding reference numbers in the revised manuscript are 41 and 42, respectively.

The manuscript has been revised as follows in P5, lines 85-87, line 99, and P6 lines 103-104, respectively:

Lines 85-87: "Second, the acoustic impedance of the cells (typical value is between $1.6\text{--}1.8 \times 10^6$ rayls) is very close to that of the medium (for example, the values of water and soft tissue average are 1.5×10^6 , and 1.58×10^6 rayls, respectively) ^{41,42}".

Line 99: "the acoustic contrast has increased by nearly 15 times".

P6, lines 103-104: "The maximum simulated ARF exerted on GVs@*E. coli* is about 63-fold higher than that of the control bacteria without GVs".

Reference:

[1] Fonseca, A. D. C., Kohler, T., & Ahmed, D. (2022). Ultrasound - Controlled Swarmbots Under Physiological Flow Conditions. *Advanced Materials Interfaces*, 9(26), 2200877.

[2] Augustsson, P., et al. Iso-acoustic focusing of cells for size-insensitive acousto-mechanical phenotyping. *Nature Communications* 7, 2016.

2) When the GVs are described, line 89 it would be good to mention their size.

Response:

Thanks for reviewer's constructive comments. We have the typical size of the GVs, which are usually approximately rod-shaped structures with a length of 100-600 nm and a width of 40-200 nm.

The manuscript has been revised as follows in P5, lines 91 to 92:

“GVs, which are usually approximately rod-shaped structures with a length of 100-600 nm and a width of 40-200 nm”.

3) P6 it would be good to comment that the acoustic field shape was simulated (my understanding, please correct if wrong).

Response:

Thanks for reviewer's good suggestions. We have added the explanation that the 3D acoustic field displayed in Fig. 2f is based on simulation data. The corresponding measurement data was also provided in Supplementary Fig. 2c.

The manuscript has been revised as follows in P7, line 128 and line 132:

Line 128: “As shown in simulated 3D acoustic field (Fig. 2f), a tiny focal beam (focal depth: 6 mm, f-number = 1) could be generated by exciting the array with an iterative backpropagation (IB) algorithm”.

Line 132: The phrase “Fig.2f and” has been deleted.

4) P7, line 129 (and line 286) it is hypothesised that the main competing force for the bacteria is Brownian motion. I'd have thought it was flow induced drag? Some evidence of the Brownian motion hypothesis is needed.

Response:

Thanks for reviewer's insightful comments. As the reviewer thought, our previous statements about competing force for the bacteria are inaccurate. In the absence of ultrasound, the random Brownian motion produced by water molecules continuously hitting bacteria at random mainly determines the motion state of individual bacterium. However, when the ultrasound is turned on, the motion state of tiny bacteria under the action of acoustic field is determined by both the acoustic radiation force (including primary and secondary acoustic radiation force) and the Stokes force caused by the acoustic streaming. For micrometer scale common bacteria without GVs, the motion state is significantly affected by Stokes force, owing to the radiant force is proportional to the radius of the target, while the Stokes force is proportional to the radius of the target. And owing to the positive acoustic contrast (+0.07) of common bacteria, the direction of the radiant force exerted on them in the focal spots is away from the focal center. As a result, common bacteria will be slowly expelled from the focal spots by the combined effects of radiation force and acoustic streaming. However, for GVs@*E. coli*, the presence of GVs significantly decreases the average density and increases the compressibility, resulting in the acoustic contrast of the genetically engineered bacteria changing from +0.07 to -1.1, flipping the sign of the acoustic contrast from positive to negative, and increasing its magnitude by

approximately 15 times. The direction of the radiant force exerted on the GVs@*E. coli* is changed to point towards the focal center. And the magnitude of the radiant force increases greatly enough (the maximum simulated acoustic radiation force exerted on GVs@*E. coli* is about 63-fold than that of the control bacteria without GVs) to overcome Stokes force. In addition, when multiple GVs@*E. coli* migrate close to each other, cross-scattering of these reflections from these GVs@*E. coli* generates a localized pressure gradient and gives rise to secondary radiation forces, which act on adjacent GVs@*E. coli*, facilitating the GVs@*E. coli* to form a cluster [1]. This secondary radiation force, which is attractive, is mainly caused by the in-phase oscillation of GVs of adjacent bacteria. As a result, GVs@*E. coli* will gather towards the focal center dominated by primary and secondary radiant forces overcoming the barrier of Stokes force. We have replaced the statement about Brownian motion with Stokes force and added the reference [2] (the corresponding reference number in the manuscript is 47) that introduces the microparticle acoustophoresis driven by acoustic radiation forces and Stokes force caused by the acoustic streaming.

The manuscript has been revised as follows in P7, lines 142 to 143, P8, line 149 and P16, line 328:

P7, lines 142 to 143,: “Thus, the control bacteria without GVs, which possess positive acoustic contrast, are directed away from the focus by the ARF (Fig. 2g) and are too weak to overcome Stokes force caused by the acoustic streaming⁴⁷.”

P8, line 149: “This strong ARF with the direction pointing to the focal center will help these genetically engineered bacteria with GVs to overcome Stokes force.”

P16, line 328: "... these genetically engineered bacteria overcome Stokes force"

Reference:

[1] Fonseca, A. D. C., Kohler, T., & Ahmed, D. (2022). Ultrasound - Controlled Swarmbots Under Physiological Flow Conditions. *Advanced Materials Interfaces*, 9(26), 2200877.

[2] Muller, P. B., Barnkob, R., Jensen, M. J. H., & Bruus, H. (2012). A numerical study of microparticle acoustophoresis driven by acoustic radiation forces and streaming-induced drag forces. *Lab on a Chip*, 12(22), 4617-4627.

5) Around this same region it would be useful to explain that the mechanism of trapping is one of changing the local average mass density. This would make it clear that the action is a combined effect of many GVs, rather than individual GVs.

Response:

Thanks for reviewer's constructive comments. We have added some explanations about the mechanism of trapping GVs@*E. coli*, including the presence of GVs significantly decreases the average density and increases the compressibility and introduces in-phase oscillation of GVs of adjacent bacteria, leading the bacteria to gather towards the focal center dominated by primary and secondary radiant forces overcoming the barrier of Stokes force. We have added the reference [1] (the corresponding reference number in the revised manuscript is 41) that explains the mechanism of cluster formation promoted by in-phase vibration of the adjacent bubbles.

The manuscript has been revised as follows in P8, lines 150 to 153, and P16, lines 324 to 329:

P8, lines 150 to 153: “Moreover, when multiple GVs@*E. coli* are close to the focal center, cross-scattering of these reflections from GVs generates a localized pressure gradient and gives rise to secondary radiation forces which attract adjacent bacteria (mainly caused by the in-phase oscillation of GVs), and facilitate the GVs@*E. coli* to form a cluster⁴¹.”.

P16, lines 324 to 329: “The presence of GVs significantly reduced bacterial average density and increased compressibility, thus greatly enhancing the sensitivity of the whole bacterium to ultrasound and increasing tens of times ARF exerted on the bacterium. Moreover, the in-phase oscillation of GVs of adjacent bacteria introduced significantly increased secondary radiation force, which also facilitated GVs@*E. coli* to aggregate. As a result, these genetically engineered bacteria can overcome Stokes force and be trapped or driven by the ARF.”

Reference:

[1] Fonseca, A. D. C., Kohler, T., & Ahmed, D. (2022). Ultrasound - Controlled Swarmbots Under Physiological Flow Conditions. *Advanced Materials Interfaces*, 9(26), 2200877

6) P10, line 190 onwards, describes the use of a vortex to rotate the clusters. Can the authors comment on the purpose of this aspect? Are there applications that could benefit from this, or are they just demonstrating an interesting capability?

Response:

Thanks for reviewer's question. Herein, we choose GVs@*E. coli* as probes for characterizing the transformation of the acoustic fields, based on their characteristic of clustering in areas of high acoustic intensity. When the preset topological charge m changes from 1 to 3, the patterns of the vortex fields displayed by the bacterial cluster change accordingly. This GVs-enabled cell clustering ability may facilitate lots of potential biomedical applications such as 3D cell assembly, tissue engineering and mechanobiology [1]. In addition, the rotation of cells or the cell clusters caused by vortices can facilitates single-cell analysis [2], drug discovery [3], and cell diagnosis [4]. The hidden structural, cellular, and genetic details may be revealed by rotation, which are of great significance in small organism phenotyping [5], 3D mechanical characterization [6], and screening [7]. In our study, besides displaying the transformation of the acoustic fields, rotation of the clusters may also be used to redirect the cluster in the bifurcation of blood vessels, and the shear force of vortices may be used to increase the permeability of endothelial cell space and promote the homing of bacteria to the tumour site.

Reference:

[1] Ma, Z., Holle, A. W., Melde, K., Qiu, T., Poeppel, K., Kadiri, V. M., & Fischer, P. (2020). Acoustic holographic cell patterning in a biocompatible hydrogel. *Advanced Materials*, 32(4), 1904181.

[2] Pushkarsky, I., Liu, Y., Weaver, W., Su, T. W., Mudanyali, O., Ozcan, A., & Di Carlo, D. (2014). Automated single-cell motility analysis on a chip using lensfree microscopy. *Scientific Reports*, 4(1), 1-9.

[3] Chung, A. J., Huh, Y. S., & Erickson, D. (2009). A robust, electrochemically driven microwell drug delivery system for controlled vasopressin release. *Biomedical Microdevices*, 11(4), 861-867.

[4] Elbez, R., McNaughton, B. H., Patel, L., Pienta, K. J., & Kopelman, R. (2011). Nanoparticle induced cell magneto-rotation: monitoring morphology, stress and drug sensitivity of a suspended single cancer cell. *PloS One*, 6(12), e28475.

[5] Ahmed, D., Ozcelik, A., Bojanala, N., Nama, N., Upadhyay, A., Chen, Y., ... & Huang, T. J. (2016). Rotational manipulation of single cells and organisms using acoustic waves. *Nature Communications*, 7(1), 1-11.

[6] Läubli, N. F., Burri, J. T., Marquard, J., Vogler, H., Mosca, G., Vertti-Quintero, N., ... & Nelson, B. J. (2021). 3D mechanical characterization of single cells and small organisms using acoustic manipulation and force microscopy. *Nature Communications*, 12(1), 1-11.

[7] Pardo-Martin, C., Chang, T. Y., Koo, B. K., Gilleland, C. L., Wasserman, S. C., & Yanik, M. F. (2010). High-throughput in vivo vertebrate screening. *Nature Methods*, 7(8), 634-636.

7) P11, line 208, it would be good to estimate how many GVs are in a typical cluster.

Response:

Thanks for reviewer's constructive comments. In order to estimate how many GVs are in a typical cluster, we measured the total number of bacteria in different cluster, when bacterial concentration was increased ($OD_{600}=0.5, 1, 2$), the number of bacteria in the cluster also varied, with the average number was about 3.74×10^5 CFU, 5.95×10^5 CFU, 7.88×10^5 CFU, respectively (Figure 1a and 1b in this response letter). Also, we counted the amount of GVs in a single bacterium (Figure 1c), with the number range from 245 to 489 and average number ($n=4$) was 382. As a result, we know that the GVs in different clusters are 1.43×10^8 , 2.27×10^8 , 3.01×10^8 CFU, respectively. Unfortunately, GVs are not always uniformly and regularly distributed in bacteria and they often overlap, which leads to some error in counting GVs. We have added some estimations about how many GVs are in a typical cluster and added Figure 1 to the supplementary information as Supplementary Fig 3.

Figure 1. **a**, The number of bacteria in the cluster, when ultrasound effected on different concentrations of bacteria ($OD_{600} = 0.5, 1, 2$). **b**, Quantification of (a). **c**, Transmission electron microscope image of GVs@*E. coli*. Scale bar = 500 nm.

The manuscript has been revised as follows in P8, lines 159-161:

“There are about $3 \sim 8 \times 10^5$ CFU in a typical cluster of GVs@*E. coli* (Supplementary Fig. 3a – 3b) and each bacterium has about 200 ~ 500 GVs (Supplementary Fig. 3c – 3d), forming about $1 \sim 3 \times 10^8$ GVs in a typical cluster of GVs@*E. coli*.”.

8) P 19, line 383, how is it known that the microbubbles occupy 10% of the space? This seems to be quite speculative. Hence it might be better to give a range of plausible values and quote upper and lower limits.

Response:

Thanks for reviewer’s constructive comments. As shown in Fig.1c and 1d in this response letter, we counted the amount of GVs in a single bacterium, with the number range from 245 to 489 (n=4). We assume that a single GV is 100 nm long and 50 nm wide, and that a single bacterium is 3 μ m long and 0.5 μ m wide. GV and bacterium are assumed as cubes. So we can figure out that GVs occupy 8 % to 16 % of the volume of a single bacterium.

The manuscript has been revised as follows in P21, line 431:

“By assuming that GVs occupy 10% of the intracellular space (typically, GVs occupy 8 % to 16 % of the volume of a single bacterium)”.

9) In figure 4 and 5 it would be helpful to mark the focal point. Do the circles in fig 5 represent the focal zone? It would be good if they did and in any case, it would be helpful to have this information non fig 4 as well.

Response:

Thanks for reviewer's insightful comments. We have marked the focal point in Fig.4 and Fig.5. The white circle in Fig. 5 represents the focal zone. We have also modified Fig. 4 to include white circles representing the focal zones and some quantitative presentations of the experimental results.

The modified Fig. 4 in the manuscript:

Fig. 4 | *In-vivo* acoustic trapping of GV@E. coli. **a**, Schematic diagram of *in-vivo* experiment setup. I and II are the schematic diagrams of the acoustic trapping processes of the GV@E. coli and control E. coli, respectively. **b**, **c**, Comparison of acoustic trapping of GV@E. coli (**b**) and control E. coli (**c**) in superficial blood vessels on the backs of mice. I in (**b**) and (**c**) indicates the microscopic images of blood vessels in the absence of ultrasound after injection of GV@E. coli and control E. coli, respectively. II in (**b**) and (**c**) shows the microscopic images of blood vessels after exposure to ultrasound for 180 s based on situation I in (**b**) and (**c**), respectively. Only GV@E. coli can be trapped at the focal beam centre and form clusters in the vessels. **d**, Acoustic trapping of GV@E. coli in blood vessels of different diameters. I, II, III, and IV are microscopic images of 110-, 130-, 170-, and 200- μm -diameter vessels injected with GV@E. coli before ultrasound is on, respectively. V–VIII are microscopic images

of trapped GVs@*E. coli* clusters in the corresponding blood vessels in cases I–IV after ultrasound is turned on for 20 s. The yellow arrows, white dotted circles, symbol d , t , and v in (b) and (c) indicate the blood flow direction, focal zones, vessel diameter, time and flow velocity, respectively. e, The curves of normalized fluorescence intensity increment over time within the focal zones under vessels of different diameters in (d). $t1$ to $t4$ represent the moment when the maximum increment of fluorescence intensity is reached in 200-, 170-, 130-, 110- μm diameter vessels, respectively. f, The area of the GVs@*E. coli* cluster in vessels of different diameters (after ultrasound is turned on for 20 s). The *in-vivo* acoustic trapping of GVs@*E. coli* is shown in Movie S2.

The corresponding quantitative description of the experimental results is added in P11, line 228 and P12, lines 230 to 231:

P11, line 228: "...diameters from 110 to 200 μm ."

P12, lines 230 to 231: "The GVs@*E. coli* cluster formation time decreased (Fig. 4e) and cluster size increased (Fig. 4f) with increasing blood vessel diameter..."

Reviewer #2 (Remarks to the Author):

The manuscript by Yang et al describes the *in vivo* manipulation of gas vesicle expressing *E. Coli* using a programmable acoustic array. Building on previous findings that GV expression in bacteria allows them to experience strong acoustic radiation force, this study shows that such bacteria can be manipulated with an electronically programmable array, that such manipulation works within the vasculature in mice, and that this allows systemically injected bacteria to be acutely concentrated in subcutaneous tumours. This is an interesting study that merits consideration.

On the question of significance, reference 41 (Wu et al) already introduced the concept of GV-based acoustic manipulation and demonstrated that GV-expressing *E. Coli* can be selectively manipulated by ultrasound *in vitro*, including being trapped at an acoustic focus and moved around in solution. As far as I can tell, Yang et al also used the same bacterial genetic constructs that were used in that study. The authors misrepresent the contents of ref 41 by citing it as an “imaging” study in the introduction on line 89. They reference this study again on line 128 when discussing the acoustic contrast factor of GV-expressing *E. Coli*, but again in a way that obscures its key results. I think the authors should modify their introduction to more honestly cite and describe this study, from which they clearly drew inspiration for their own work.

Having said that, I don't view the existence of Wu et al as a total deal-breaker for the publication of this manuscript because this new work advances into acoustic

manipulation of bacteria *in vivo*, which was not previously demonstrated. However, I think the *in vivo* demonstration of this technique and its utility needs to go considerably further for it to be suitable for a journal like Nat Comms. The basic demonstration that focused ultrasound can trap and move cells inside blood vessels is not a sufficiently large step on its own, in my view, because it is straightforwardly expected from *in vitro* results. On the other hand, the application of acoustic trapping to enhancing tumour treatment would be sufficiently more significant. However, I think this demonstration needs to be much more convincing.

As of now, the tumour demonstration only shows that acutely after cell administration, focused ultrasound can concentrate the bacteria in tumours. However, whether this very short-term effect would have any impact on anti-tumour probiotic therapy is uncertain. tumour-homing bacteria have a well-established ability to colonize tumours on their own, and are thought to initially penetrate in small numbers and rapidly proliferate through cell division within the immunosuppressed regions of the tumour. While aggregating the bacteria at the tumour site at the time of administration could potentially give this process a boost, this is not necessarily the case. For example, bacteria concentrated in major vessels may not enter the correct niche inside the tumour, and aggregation could conceivably even be harmful to their ability to do so. Furthermore, having a large local bolus of bacteria could stimulate the immune system against the bacterial agent, reducing its long-term persistence. To demonstrate that the *in vivo* manipulation of bacterial agents is actually beneficial in tumour treatment, it would be necessary to measure bacterial colonization over several weeks

with and without ultrasound and monitor its effects on tumour growth and animal survival. This is standard in other tumour-homing bacteria studies (e.g. see studies by Tal Danino or Jung-Joon Min). Furthermore, I think it is essential to include controls for ultrasound treatment combined with non-GV-expressing *E. Coli* because ultrasound by itself could have a damaging or immunogenic effect on the tumour (e.g. through heating).

Bacterial manipulation using a programmable array is also a new contribution relative to the previous use of single-element transducers and standing waves. However, since people have used such arrays to manipulate other particles, I don't think this by itself is a sufficient advance to compensate for insufficient *in vivo* data.

Response:

Thanks for reviewer's insightful comments. We agree with you that our work has been inspired by *Reference 41* (Wu., et al). Wu. et al have already introduced the concept of GV-based acoustic manipulation and demonstrated that GV-expressing *E. coli* can be selectively manipulated by ultrasound *in vitro*, including being trapped at an acoustic focus and translated around in solution by moving the transducer. Based on the work of Wu. et al, our research aimed to realize the acoustic manipulation of GV-expressing *E. coli* by using the holographic acoustic tweezers technology *in vivo* and further validate the therapeutic potential of our approach for tumours. There are many challenges with *in-vivo* acoustic manipulation: 1) Complex *in-vivo* tissue structures and normal blood flow environment require flexible, precise 3D manipulation abilities of acoustic tweezers. 2) Non transparent tissues make it

difficult to observe the acoustic manipulation situation. 3) Heterogeneous tissue may cause distortion of the acoustic fields. In addition, the small size (typically only a few microns) of cells and their close acoustic impedance to the medium (typical value is between $1.6\text{--}1.8 \times 10^6$ rayls compared to 1.5×10^6 rayls that of water) make it even more difficult for *in-vivo* acoustic manipulation of cells. In this study, we have successfully demonstrated the acoustic manipulation of the bacteria (whose ultrasound sensitivity was enhanced by the GVs) in mouse blood vessels. The array-based holographic acoustic tweezers, which have a very powerful ability to programmatically adjust the waveform, amplitude and phase information of each element of the array, can form various complex acoustic fields and multiplex these fields in time. They allow trapping the targets in the acoustic traps and moving the targets in the three-dimensional space without physical movement of the transducer. The acoustic fields also can be transformed flexibly and thus various particles with different acoustic contrast coefficients can be trapped (the negative or positive acoustic contrast particles can be trapped by multi-foci or the vortex field, respectively). Herein, the GVs@*E. coli* were firstly trapped, translated, rotated and used for displaying the acoustic field transformation by using the holographic acoustic tweezers *in vitro*. And then, we extended the *in-vivo* acoustic manipulation of the GVs@*E. coli*. The single-side-array-based holographic acoustic tweezers have been proved to be one of the most suitable methods for *in-vivo* manipulation due to their flexible experimental settings and powerful manipulation capabilities. They have facilitated the *in-vivo* manipulation for non-living objects, such as the manipulation of

glass spheres in a pig bladder and the trapping of microbubbles in mouse vessels, while their potentials for *in-vivo* acoustic manipulation of cells has not yet been reported. So, the *in-vivo* acoustic manipulation of cells reported in this paper, which was realized in a challenging environment with normal blood flow, will greatly promote the development of acoustically manipulated targeted drug delivery, tissue repair and reconstruction and other *in-vivo* applications. In addition, the flexible signal modulation capability of the array can realize the interval emission between imaging pulse and manipulation pulse, which may facilitate acoustic manipulation of cells *in vivo*, especially in deep tissues, under the guidance of ultrasonic image in the future.

As for citing of *Reference 41* (Wu., et al), we have revised the manuscript according to the suggestions of reviewer as follows in P5, lines 93 to 97:

“enabling their imaging^{43,44} and *in-vitro* manipulated⁴⁵ by ultrasound. In this study, inspired by these previous work, our study aimed to realize the acoustic manipulation of GV-expressing bacteria by using the holographic acoustic tweezers technology *in vivo* and further to validate the therapeutic potential of our approach for tumours. In details,”. Here, *Reference 45* is *Reference 41* (Wu., et al) before revised.

In addition to manipulation, as the reviewer suggested, we further demonstrated in the revised manuscript that trapping GVs@*E. coli* at the tumour site using holographic acoustic tweezers can significantly improve the migration and resettlement of bacteria at the tumour site and effectively inhibit the tumour growth, which will provide new

ideas and methods for improving the efficacy of cellular immunotherapy for tumour treatment.

According to the reviewer's kind suggestions, we added the relative data about the tumour treatment and bacterial colonization in tumour. In the early stage, we selected YB1 strain with good anti-tumour effect for subsequent studies [1]. Firstly, YB1 was transformed with the pET28a_T7-ARG1 plasmid to obtain GV_s@YB1 that could express GV_s [2]. After establishing the tumour-bearing mice, these mice were divided into five groups, including Control, US, *E. coli* + US, GV_s@YB1 and GV_s@YB1 + US. When the tumour volume reached about 50 mm³, the bacteria were intravenously administrated and the tumours were received with ultrasonic irradiation. The tumour volume and survival rate of mice were recorded (Figure 2a). The results showed that there were not significant tumour inhibitory effects in the Control and US groups. Only slight tumour inhibitory effects were observed in the *E. coli* + US and GV_s@YB1 groups. By contrast, GV_s@YB1 + US group exhibited the strongest tumour inhibitory effect (Figure 2b and 2e). In addition, the survival time of mice in the GV_s@YB1 + US group showed significantly longer than other groups (Figure 2d). These data indicated that the GV_s@YB1 + US group could achieve stronger tumour growth inhibition effect and significantly lengthen the survival time of tumour-bearing mice.

Figure 2. Acoustic aggregation of GV@YB1 improves the anti-tumour efficacy. a, Schematic illustration of acoustic aggregation of GV@YB1 to inhibit tumour growth in tumour-bearing mice. b, tumour growth curves of tumour-bearing mice in different treatment groups. c, Body weight change curve of tumour-bearing mice in different treatment groups. d, Survival curves for different treatment groups. e, Representative photos of tumour growth in each group of tumour-bearing mice after different treatments.

In addition, we evaluated bacterial colonization in tumour. The tumour-bearing mice were injected intravenously with 5×10^7 CFU GV@YB1, and received with acoustic irradiation for 20 min. The tumour tissues were collected and homogenized

after 7 or 14 days. The diluted tissue homogenate was plated and cultivated on the LB agar plates with 100 $\mu\text{g}/\text{mL}$ kanamycin. The results showed that GV@YB1+ US group had obviously more bacteria in the tumour than that of GV@YB1 group after 7 and 14 days (Figure 3a and 3b). These data demonstrated that acoustic manipulation could increase the aggregation of GV@YB1 in the tumour and keep their acoustic manipulation up to two weeks.

Figure 3. Acoustic aggregation of GV@YB1 in tumour. a, the number of GV@YB1 in the tumour of tumour-bearing mice with or without ultrasonic irradiation, at 7 and 14 days after intravenous administration of 5×10^7 CFU GV@YB1. b, Quantification of (a).

As for exploring the therapeutic effect on tumour, we added Figure 2 to the revised manuscript as Fig. 7 and Figure 3 to the supplementary information as Supplementary Fig. 8. In addition, we added the relative data and statements in the revised manuscript according to the suggestions of reviewer as follows in P6, lines 107 to 109, P15, lines 295 to 313, P18, line 370, P19, lines 378 to 382, P23, lines 460 to 473, and P24, lines 495 to 496:

P6, lines 107 to 109: “Moreover, we further demonstrated that trapping GV@YB1 at the tumour site using holographic acoustic tweezers can significantly improve the

migration and resettlement of bacteria at the tumour site and effectively slow down the growth rate of the tumour.”

P15, lines 295 to 313:

“Acoustic aggregation of GVs@YB1 for tumour therapy

Next, we further evaluated the anti-tumour efficacy of acoustic manipulation of GVs@YB1 *in vivo*. The treatment strategy for tumour-bearing mice was shown in Fig. 7a. The tumour-bearing mice were randomly divided into five groups, including PBS (Control), ultrasound alone (US), ultrasound combined with *E. coli* BL21 without GVs (*E. coli* + US), YB1 with GVs (GVs@YB1), and GVs@YB1 combined with ultrasound (GVs@YB1 + US). As shown in Fig. 7b and 7e, no significant tumour inhibitory effects were found in the US group, reaching more than 1500 mm³ mean tumour volume at day 17, similar with the Control group. Slight tumour inhibitory effects were observed in the *E. coli* + US and GVs@YB1 groups, with 1350 mm³ and 800 mm³ mean tumour volume at day 21, respectively. By contrast, GVs@YB1 + US group exhibited the strongest tumour inhibitory effect, with only 360 mm³ mean tumour volume at day 21. No significant body weight changes were observed in these tumour-bearing mice during the treatment period (Fig. 7c). Notably, the GVs@YB1 + US group exhibited longest survival time, achieving more than 45 days in comparison with other groups (Fig. 7d). These data indicated that the GVs@YB1 + US group could more effectively inhibit tumour growth and improve the survival time of tumour-bearing mice. Furthermore, we detected the GVs@YB1 aggregation in tumour of tumour-bearing mice at day 7 and day 14 after treatment. Our results

showed that the number of GV@YB1 in the tumour of GV@YB1 + US group was significantly more than that in GV@YB1 group (Supplementary Fig. 8).”

P18, line 370: “**Preparation of GV@*E. coli* and GV@YB1**”

P19, lines 378 to 382: “For GV@YB1, the pDT103 (Addgene plasmid # 106475) was transformed into *salmonella Typhimurium* YB1 (YB1) by using of the same protocol. The culture process of transformed YB1 was the same with *E. coli* BL21, except that 3 nM N-(β -ketocaproyl)-l-homoserine lactone (AHL, sigma # K0037) was added when the concentration of YB1 culture achieved 0.3 OD_{600nm} to induce the expression of GVs.”

P23, lines 460 to 473:

“**Establishment of tumour-bearing mouse model and tumour treatment**”

Female Balb/c mice aged 4-6 weeks were selected to establish breast tumour model. Briefly, Mice were kept in a Specific-pathogen-free (SPF) environment at 23-25°C on a 12-h light/dark cycle and provided with adequate water and food. 100 μ l PBS solution containing 5×10^5 4T1 breast cancer cells was injected subcutaneously into the right thigh of mice. When the tumour volume reached 50 mm³, these tumour-bearing mice were randomly divided into five groups, including control group (intravenous injection of 100 μ l PBS), US group (ultrasound irradiation for 20 min), *E.coli* + US group (intravenous injection of 5×10^7 CFU *E. coli* BL21 without GVs and then ultrasound irradiation for 20 min), GV@YB1 group (intravenous injection of 5×10^7 CFU GV@YB1), GV@YB1 + US group (intravenous injection of 5×10^7 CFU GV@YB1 and then ultrasound irradiation for 20 min). Acoustic parameters

were set at 20 V input voltage and 10 % duty cycle. The tumour volume was measured every two days. When the tumour volume exceeded 2000 mm³ or the experiment ended, these tumour-bearing mice were euthanized by carbon dioxide asphyxia.”

P24, lines 495 to 496:

“The results were significant when ****P < 0.0001; ***P < 0.001; **P < 0.01; *P < 0.05.”

Reference:

[1] Chen F.M., et al. Nanophotosensitizer-engineered Salmonella bacteria with hypoxia targeting and photothermal-assisted mutual bioaccumulation for solid tumour therapy. *Biomaterials*. 214, 119226 (2019).

[2] Bourdeau, R. W., et al. Acoustic reporter genes for noninvasive imaging of microorganisms in mammalian 538 hosts. *Nature* 553, 86-90 (2018).

For minor points, the detailed responses are given in a point-by-point manner below and appropriate changes have been made in accordance with the reviewer’ s suggestions. In the revised manuscript, revisions have been highlighted in yellow.

Thanks again to the reviewer for insightful suggestions and questions, which are of great help to the improvement of our manuscript.

Some minor points:

How do the authors explain the concentration dependence of cell cluster size in Fig. 2n? Are the clusters large enough that they take up all the bacteria available, is it a kinetic effect (in which case it would be interesting to see what happens after longer exposure), or are there some inter-cell acoustic or chemical interactions?

Response:

Thanks for reviewer's constructive comments. As the reviewer thought, the process of acoustic aggregation of GVs@*E.coli* is a kinetic progress, which the bacteria first migrate to the focal center under the dominant of the primary radiation force, and then form a cluster by multiple neighboring bacteria under the dominant of the secondary radiation force. The whole process needs to overcome the Stokes force caused by the acoustic streaming. And not all the available bacteria will be trapped by the focal acoustic field. As show in the Fig. 2i, only GVs@*E.coli* near the focal spots will be significantly affected by the acoustic radiation force. So when the concentration increases, that is, when there are more GVs@*E.coli* near the focus spots, more bacteria will gather under the same conditions and form larger clusters. When the bacterial concentration is very low, such as 6.25×10^7 cells/ml in this study, fewer bacteria are pushed to the center by the primary radiation force, and the average distance between two adjacent bacteria is far away, so it is difficult to form a significant secondary radiation force. As a result, the bacteria are washed away by the Stokes force caused by acoustic streaming before forming clusters.

We used ultrasound to irradiate GVs@*E.coli* for 5 min and then collected clusters to photograph under transmission electron microscopy and detect the production of

reactive oxygen species respectively. The results showed that acoustic irradiation did not cause the burst of the GVs in the bacteria (Figure 4a). Also, the fluorescence intensities of reactive oxygen species' probe in the GVs@*E.coli* + US group and GVs@*E.coli* (Control) group did not show significant difference, indicating that acoustic irradiation did not cause produce reactive oxygen species in GVs@*E.coli* (Figure 4b).

Figure 4. Structural and chemical changes after acoustic irradiation of GVs@*E.coli*. **a**, Transmission electron microscope image of GVs@*E.coli* with or without ultrasound. Scale bar = 500 nm **b**, The production of reactive oxygen species of GVs@*E.coli* with or without ultrasound.

We have added Figure 4 to the supplementary information as Supplementary Fig 4.

The manuscript has been revised as follows in P9, lines 180 to 181:

“The GVs in the bacteria will not collapse (Supplementary Fig. 4a) or cause inter-cell chemical reactions (Supplementary Fig. 4b) under these parameters”.

Fig 4d(I) appears to show bacterial cells lining the vessel walls before the ultrasound is applied. Is this so, and do the authors have an explanation for why it should be the case?

Response:

Thanks for reviewer's insightful comments. In the blood vessel shown in Fig 4d(I), we repeated the experiment of trapping bacteria for many times. The original result shown was not the state of the blood vessel before the start of the first ultrasound. It was our negligence that we started the experiment before the bacteria in the blood vessel wall was completely washed away after repeated trapping, and we have replaced the result with a new result in Fig 4. Similarly, we have replaced Fig. 4d(II) and 4d(VI) with a better experimental result.

The modified Fig. 4 in the manuscript:

Fig. 4 | *In-vivo* acoustic trapping of GV@E. coli. a, Schematic diagram of *in-vivo* experiment setup. I and II are the schematic diagrams of the acoustic trapping processes of the GV@E. coli and control E. coli, respectively. b, c, Comparison of acoustic trapping of GV@E. coli (b) and control E. coli (c) in superficial blood vessels on the backs of mice. I in (b) and (c) indicates the microscopic images of blood vessels in the absence of ultrasound after injection of GV@E. coli and control E. coli, respectively. II in (b) and (c) shows the microscopic images of blood

vessels after exposure to ultrasound for 180 s based on situation I in (b) and (c), respectively. Only GVs@*E. coli* can be trapped at the focal beam centre and form clusters in the vessels. **d**, Acoustic trapping of GVs@*E. coli* in blood vessels of different diameters. I, II, III, and IV are microscopic images of 110-, 130-, 170-, and 200- μm -diameter vessels injected with GVs@*E. coli* before ultrasound is on, respectively. V–VIII are microscopic images of trapped GVs@*E. coli* clusters in the corresponding blood vessels in cases I–IV after ultrasound is turned on for 20 s. The yellow arrows, white dotted circles, symbol d , t , and v in (b) and (c) indicate the blood flow direction, focal zones, vessel diameter, time and flow velocity, respectively. **e**, The curves of normalized fluorescence intensity increment over time within the focal zones under vessels of different diameters in (d). $t1$ to $t4$ represent the moment when the maximum increment of fluorescence intensity is reached in 200-, 170-, 130-, 110- μm diameter vessels, respectively. **f**, The area of the GVs@*E. coli* cluster in vessels of different diameters (after ultrasound is turned on for 20 s). The *in-vivo* acoustic trapping of GVs@*E. coli* is shown in Movie S2.

The corresponding quantitative description of the experimental results is added in P11, line 228 and P12, lines 230 to 231:

P11, line 228: "...diameters from 110 to 200 μm ."

P12, lines 230 to 231: "The GVs@*E. coli* cluster formation time decreased (Fig. 4e) and cluster size increased (Fig. 4f) with increasing blood vessel diameter..."

Is there a way to make the results in Fig 4 and 5 more quantitative? What is the success rate of trapping attempts? How close or far from the predicted focus do the aggregates show up? What is the size of the aggregates axially and across the vessel as a function of the vessel diameter and ultrasound parameters?

Response:

Thanks for reviewer's constructive comments. We have added quantitative presentations on the experimental results in Fig.4. Our success rate for acoustic trapping is nearly 100% at the appropriate bacterial concentration and without flow *in vitro*. However, trapping bacteria becomes difficult when the flow exists, and the maximum velocity of the flow in which we successfully trap the bacterial clusters is

4.3 mm/s. Similarly, in the *in vitro* condition without flow, the trapped bacterium clusters will appear exactly where the predicted focus to be. When the flow exists, a position deviation will present between the bacterial clusters and the predicted focus, and increase with the increase of the flow velocity. And we measured a position deviation of 413 μm at the maximum flow velocity (4.3 mm/s) that could still trap the clusters. As shown in Fig. 2l and 2m, the increase of ultrasonic parameters (including duty cycle and input voltage) mainly accelerates the formation of clusters, but has little effect on the size of clusters. The final size of the bacterial cluster mainly depends on the bacterial concentrations (Fig. 2n). The size of GVs@*E. coli* cluster increased with increasing blood vessel diameter, mainly because larger blood vessels can supply circulating bacteria more efficiently.

The authors used BL21 *E. Coli*, which lack flagella. Would acoustic trapping and manipulation work as well with motile strains, like the kind most commonly used in tumour-homing therapies?

Response:

Thanks for reviewer's insightful comments. Our holographic acoustic tweezers system also work as well with trapping and manipulating cells with flagella. Under the same parameters in this study, the biciliate, unicellular alga *Chlamydomonas reinhardtii* (CR) can be successfully been trapped and manipulated in a 10-cm-diameter Petri dish. When the ultrasound was not turned on, the CR moved freely in the Petri dish (Figure 5a). The swimming speed of the motile CR used in this work is

about 100 $\mu\text{m/s}$. When the focal acoustic field is turned on, the motile CR can be trapped and gathered into circular clusters (Figure 5b). When the acoustic field further changes from focal acoustic field to vortex or dual-focal field, the CR cluster can also display the pattern of vortex (Figure 5c) or dual-focal field (Figure 5d) correspondingly. These results suggest that our holographic acoustic tweezers system can also overcome the propulsive force introduced by the flagella to trap and manipulate motile cells, which has a bright application prospect in the motile cell based tumour-homing therapies.

Figure 5. Acoustic manipulation of CR in Petri dish. a, CR in the Petri dish before ultrasound. **b**, Trapping of CR using focal beam. **c**, Displaying the vortex acoustic field patterns ($m=1$) by using CR. **d**, Displaying the multi-foci acoustic field patterns by using CR.

Reference

- [1] Kim, M., Huff, E., Bottier, M., Dutcher, S. K., Bayly, P. V., & Meacham, J. M. (2019). Acoustic trap-and-release for rapid assessment of cell motility. *Soft Matter*, 15(21), 4266-4275.

One line 35, I think the authors mean “CAR-T” instead of “CAT-T” .

Response:

Thanks for reviewer's helpful comments. We have replaced "CAT-T" with "CAR-T".

The manuscript has been revised as follows in P3, line 37:

"including CAR-T cells".

Reviewer #3 (Remarks to the Author):

In the manuscript, the authors demonstrated the *in-vivo* manipulation of genetically engineered bacteria using electronically steered acoustic beams developed by an acoustic transducer array. The results are remarkable and the study is data-rich by comparisons between engineered and nature's *E. Coli*, as well as *in-vitro* and *in-vivo* conditions. The control trapping experiments with and without GVs inside bacteria show successful gas based acoustic navigation. The manuscript is well-organized and easy to understand. The authors showed controlled manipulation of the clusters using transducer array.

Response:

Thanks for reviewer's insightful comments. As the reviewer said, our manuscript demonstrated the *in-vivo* acoustic manipulation of genetically engineered bacteria using electronically steered acoustic beams developed by a transducer array. The GVs introduced by gene engineering enabled the engineered bacteria can be trapped and manipulated by acoustic radiation force *in-vitro* and *in-vivo* situation, while the nature *E. coli* failed to be trapped in either situation. This GVs-based manipulation may provide new methods and ideas for acoustic manipulation for cells. And lots of promising applications, including acoustically manipulated targeted drug delivery, tissue repair and reconstruction and so on, will be greatly promoted by realization of this *in-vivo* acoustic manipulation of cells. We have given the detailed responses in a point-by-point manner below and made appropriate changes in accordance with the reviewer's suggestions. In the revised manuscript, revisions have been highlighted in

yellow. Thanks again to the reviewer for insightful suggestions and questions, which greatly improved our manuscript.

Here I have some comments for you to consider:

1) “... living cells are difficult to acoustically manipulate because of their small size and the similarity between their acoustic impedance and that of the medium”

“Second, the acoustic impedance of the cells is very close to that of the medium, resulting in less ARF generation” . Citing references and quantifying the acoustic impedance of “normal cells” and “the medium” will be adequate. I suggest to cite this paper, which describes the effectiveness of microbubble manipulation in artificial vasculature (<https://doi.org/10.1002/admi.202200877>).

Response:

Thanks for reviewer’s constructive comments. We have added some quantitative data on typical acoustic impedance values for cells and medium. The acoustic impedance values are from the reviewer suggested added reference [1] and [2], which the corresponding reference numbers in the manuscript are 41 and 42, respectively.

The manuscript has been revised as follows in P5, lines 85-87, respectively:

P5, lines 85-87: “Second, the acoustic impedance of the cells (typical value is between $1.6\text{—}1.8 \times 10^6$ rayls) is very close to that of the medium (for example, the values of water and soft tissue average are 1.5×10^6 , and 1.58×10^6 rayls, respectively)^{41,42}”.

Reference:

[1] Fonseca, A. D. C., Kohler, T., & Ahmed, D. (2022). Ultrasound - Controlled Swarmbots Under Physiological Flow Conditions. *Advanced Materials Interfaces*, 9(26), 2200877.

[2] Augustsson, P., et al. Iso-acoustic focusing of cells for size-insensitive acousto-mechanical phenotyping. *Nature Communications* 7, 2016.

2) "The maximum peak-to-peak acoustic pressure in the focal zone was measured to be approximately 1.68 MPa." This level of acoustic pressure will not only destroy gas microbubbles but may induce inertial cavitation. Could the authors discuss the safety of bacteria with engineered GVs in the blood vessels? Additionally, in Figure 5a, the cluster size is significantly larger than the vasculature's diameter. Will it block the blood flow? Will it induce platelet aggregation or thrombosis?

Response:

Thanks for reviewer's constructive comments. According to the reference [1], the critical collapse acoustic pressure of the GVs (*arg1* type, same as the type in our manuscript) expressed in *E. coli* is about 2.7 MPa of the peak positive pressure, as show in Figure 3d and Extended Data Figure 5e-5f. In our study, the maximum peak-to-peak acoustic pressure in the focal zone was measured to be approximately 1.68 MPa, thus the peak positive pressure was approximately 0.84 MPa. This value (0.84 MPa) is much small than the critical collapse acoustic pressure reference value (2.7 MPa), thus the acoustic fields in our study are safe and will not cause the GVs to collapse and even induce inertial cavitation. In addition, the center frequency of the

ultrasound wave is 3 MHz and the peak rarefaction pressure is about 0.84 MPa, thus the MI value is about 0.48, which is also much less than the corresponding U.S. Food and Drug Administration (FDA) restrictions for acoustic output from diagnostic ultrasound equipment ($MI \leq 1.9$ [2]). This result also proves that the probability of inertial cavitation caused by our acoustic fields is very low.

This result is also confirmed by our experiment. As shown in Fig. 4a in this response letter, we used ultrasound to irradiate GVs@*E.coli* for 5 min and then collected clusters to photograph under transmission electron microscopy. The results show that the ultrasound did not cause the collapse of the GVs in the bacteria.

For the question whether the bacteria cluster will block the blood flow or not? Our answer is no. First, according to our experimental results and reference [3] and the reviewer suggested added reference [4], the process of manipulating tiny particles (the diameter of the particles are much smaller than the wavelength, like GVs@*E. coli* or microbubbles) in the tube (cellulose tube, blood vessel, etc.) to form clusters is a process in which particles attract each other in the transverse direction and simultaneously are migrated towards the tube wall along the wave propagation direction. Eventually, the particles are pushed and trapped against the wall of the tube and gather laterally toward the center of the acoustic trap to form clusters. The cluster we observed was clung to the tube wall and its projection is elliptical. However, the thickness of the cluster in the vertical projection direction is finite, so it is difficult to fill the entire cavity. This is also supported by our experimental results. As shown in Figure 5a and Movie S2, when the GVs@*E. coli* cluster was trapped on the wall of

the blood vessel, some of the bacteria are still flushed down the stream by the rapid blood flow near the center of the vessel, indicating that the center of the vessel is not blocked and therefore cannot introduce thrombosis.

Reference:

[1] Bourdeau, R. W., Lee-Gosselin, A., Lakshmanan, A., Farhadi, A., Kumar, S. R., Nety, S. P., & Shapiro, M. G. (2018). Acoustic reporter genes for noninvasive imaging of microorganisms in mammalian hosts. *Nature*, 553(7686), 86-90.

[2] Nelson, T. R., Fowlkes, J. B., Abramowicz, J. S., & Church, C. C. (2009). Ultrasound biosafety considerations for the practicing sonographer and sonologist.

[3] Lo, W. C., Fan, C. H., Ho, Y. J., Lin, C. W., & Yeh, C. K. (2021). Tornado-inspired acoustic vortex tweezer for trapping and manipulating microbubbles. *Proceedings of the National Academy of Sciences*, 118(4), e2023188118.

[4] Fonseca, A. D. C., Kohler, T., & Ahmed, D. (2022). Ultrasound - Controlled Swarmbots Under Physiological Flow Conditions. *Advanced Materials Interfaces*, 9(26), 2200877.

3) What is the speed of GVs in the vasculature? Can the authors characterize the velocity plot of bacteria in mice vasculature? How long does the GVs stay within bacteria? Does it grow bigger over time? The latter is not relevant to the project; however, it will be nice to consider the rectified diffusion of microbubbles/GVs inside bacteria for further projects.

Thanks very much for the reviewer's helpful comments. We have measured the velocity of the $GVs@E.coli$ in the blood vessels and plotted on the corresponding images of *in vivo* experiment.

The modified Fig. 4 in the manuscript:

Fig. 4 | *In-vivo* acoustic trapping of $GVs@E.coli$. **a**, Schematic diagram of *in-vivo* experiment setup. I and II are the schematic diagrams of the acoustic trapping processes of the $GVs@E.coli$ and control *E.coli*, respectively. **b**, **c**, Comparison of acoustic trapping of $GVs@E.coli$ (**b**) and control *E.coli* (**c**) in superficial blood vessels on the backs of mice. I in (**b**) and (**c**) indicates the microscopic images of blood vessels in the absence of ultrasound after injection of $GVs@E.coli$ and control *E.coli*, respectively. II in (**b**) and (**c**) shows the microscopic images of blood vessels after exposure to ultrasound for 180 s based on situation I in (**b**) and (**c**), respectively. Only $GVs@E.coli$ can be trapped at the focal beam centre and form clusters in the vessels. **d**, Acoustic trapping of $GVs@E.coli$ in blood vessels of different diameters. I, II, III, and IV are microscopic images of 110-, 130-, 170-, and 200- μm -diameter vessels injected with $GVs@E.coli$ before ultrasound is on, respectively. V–VIII are microscopic images of trapped $GVs@E.coli$ clusters in the corresponding blood vessels in cases I–IV after ultrasound is turned on for 20 s. The yellow arrows, white dotted circles, symbol d , t , and v in (**b**) and (**c**) indicate the blood flow direction, focal zones, vessel diameter, time and flow velocity, respectively. **e**, The curves of normalized fluorescence intensity increment over time within the focal zones under vessels of different diameters in (**d**). t_1 to t_4 represent

the moment when the maximum increment of fluorescence intensity is reached in 200-, 170-, 130-, 110- μm diameter vessels, respectively. **f**, The area of the GV@*E. coli* cluster in vessels of different diameters (after ultrasound is turned on for 20 s). The *in-vivo* acoustic trapping of GV@*E. coli* is shown in Movie S2.

The modified Fig. 5 in the manuscript:

Fig. 5 | *In-vivo* programmable acoustic manipulation of GV@*E. coli*. **a**, Manipulation of the GV@*E. coli* cluster to move back and forth along the blood vessel, while making the cluster stop moving for 30 s at specific node positions. **b**, Manipulating the GV@*E. coli* cluster selectively through the fork of the vessels. **c**, Simultaneously manipulating double clusters in the same blood vessel and driving the clusters to move closer or farther away from each other. I in (a)–(c): Schematics of manipulation experiment setups. I–IX in (a)–(c): Time-lapse microscopic fluorescence image sequences of the locations of the manipulated GV@*E. coli* clusters in each case. The yellow arrows, green arrows, white dotted circles, symbol d , t , and v in (a), (b), and (c) indicate the blood flow direction, translation directions of the clusters, focal zones, vessel diameter, time and flow velocity, respectively. The *in-vivo* programmable acoustic manipulation of GV@*E. coli* is shown in Movie S2.

The time of GVs stay within bacteria was evaluated by continuous observation of the bacteria floating and detection of intracellular GVs with transmission electron microscopy (TEM). The results showed that there was no significant change in the bacteria from the first day to the ninth day (Figure 6a), and the electron microscope results showed that the amount and size of GVs in the bacteria on the first day were not significantly different from that on the ninth day (Figure 6b). Therefore, GVs could stay in the bacteria more than ninth day and the size of GVs had not significantly change.

Figure 6. Stability assessment of GVs in *E. coli* BL-21. **a**, Floatage of GV@*E. coli* on 1, 3, 5, 7, 9 days, The bacteria were cultured into LB solution. **b**, Transmission electron microscope images of GV@*E. coli* on 1 day and 9 day respectively. Scale bar = 500 nm.

4) “ However, further increasing the duty cycle from 7% to 10% did not significantly decrease the formation time (Fig. 2m)” why? The acoustic intensity is proportional to the input.

Response:

Thanks for reviewer’s insightful question. When the bacteria haven’t yet formed clusters, the motion state of tiny bacteria under the action of acoustic field is determined by both the primary acoustic radiation force and the Stokes force caused by the acoustic streaming. According to Newton's second law, the velocity increment (Δv) of the manipulated GVs@*E. coli* during each pulse period T can be written as,

$$\Delta v = \frac{1}{m_p} \int_0^D [F_{rad} - 6\pi\eta vr] dt + \frac{1}{m_p} \int_D^T -6\pi\eta vr dt \quad (1)$$

where m_p is the mass of the cell, $[0, D]$ is the on-time duration of the whole T , F_{rad} is the primary acoustic radiation force and $6\pi\eta vr$ denotes the Stokes force when Reynolds number is less than 0.1. In detail, η is the dynamic viscosity, r is the radius of the cell and v is its velocity. It can be seen from equation (1) that under the action of acoustic field in one pulse period, the motion state of the cell is a process of first acceleration and then deceleration. However, with successive pulses, the cell's velocity does not always add up. When the cell is just starting to move, F_{rad} is much greater than the Stokes force because the cell's velocity is quite slow. The cell's velocity will continue to increase under the radiation force and the viscous effect of the liquid has little effect on the acceleration. When accelerated for a period of time, the cell velocity increases to a point where the viscosity of the fluid increases and the

cell is balanced by acoustic radiation force and Stokes force. In this case, the mean value of cell velocity in one pulse period is almost constant (the critical velocity, v_c , depends mainly on the acoustic radiation force and liquid viscosity), and the cell behaves as approximately uniform motion on the macro level. Therefore, under the action of continuous pulsed acoustic fields, cells are in a process of first acceleration and then uniform movement. When the duty cycle is increased, the velocity increment (Δv) in each pulse increases, and cells can reach the critical speed in a shorter time for uniform movement. So, when the duty cycle is increased to a certain value (such as 7% in this paper), the time for cells to accelerate to the critical speed is very short, and they mainly move at the critical speed. At this time, further increasing the duty cycle (such as, from 7% to 10%) does not significantly decrease the formation time of the cluster.

5) “The bacterial cluster is not generated at a concentration of 6.25×10^7 cells/ml” why lower concentration cannot result in aggregation? Also, is it a limitation the fact that you need higher bacterial concentrations to see clustering? What concentrations of bacteria would be toxic for the animal?

Response:

Thanks for reviewer’s constructive question. The concentration of GVs@*E. coli* towards the focal center under the acoustic field is the result of the combined action of acoustic radiation force (including primary and secondary acoustic radiation force) and Stokes force caused by the streaming. The primary and secondary radiant forces

will promote the GVs@*E. coli* to gather towards the focal center, while Stokes force will impede this process. In detail, as the force analysis shown in Fig. 2i in the manuscript, although the direction of the primary acoustic radiation force exerted on the cells near the focal zone all points to the focal center, its amplitude is not the same, but proportional to the gradient of acoustic pressure. Therefore, the maximum primary acoustic radiant force occurs at the position where the acoustic pressure changes the most, while the primary acoustic radiant force at the focal center is 0. After the GVs@*E. coli* are pushed toward the focal center by the primary acoustic radiant force, multiple bacteria will move closer to each other. Cross-scattering of these reflections from multiple GVs@*E. coli* generates a localized pressure gradient and gives rise to secondary radiation forces, which act on adjacent GVs@*E. coli*, leading to GVs@*E. coli* self-attraction [1]. This secondary radiation force, which is attractive, is mainly caused by the in-phase oscillation of GVs of adjacent bacteria. As the GVs of neighbouring bacteria were approximately equal in size and were activated by the same frequency, they oscillate in phase and further promote the bacterial self-assembly. When the GVs@*E. coli* are very close to the focal center, the primary radiation force decreases to close to 0, and the secondary radiation force dominates, resulting in the formation of the final clusters. When the bacterial concentration is very low, such as 6.25×10^7 cells/ml in this study, fewer bacteria are pushed to the center by the primary radiation force, and the average distance between two adjacent bacteria is far away, so it is difficult to form a significant secondary radiation force. As a result, the bacteria are washed away by the Stokes force caused by acoustic

streaming before forming clusters. Thus, as the reviewer said, this is a limitation that a certain concentration of bacteria (above 12.5×10^7 cells/ml in a 500 μm inner diameter silica tube in this study) is needed to generate significant secondary radiant forces at the focal center and thus form clusters. In *in-vivo* experiment, GVs@*E. coli* (concentration: 200×10^7 CFU/ml, dose: 125 μL each time, thus 25×10^7 bacteria in total) were systemically administrated by tail vein injection. A 25 g adult mouse has about 1.8 ml blood . So the concentration of bacteria in the mouse circulatory system is about 13.8×10^7 CFU/ml, which is above the critical concentration 12.5×10^7 CFU/ml of *in-vitro* experiment, resulting in the successful aggregation of the bacteria *in vivo*. In addition, biological safety of our method was also verified by experiments. We did the haematological examination and the histological analysis (H&E staining) with GVs@*E. coli* + US group, GVs@*E. coli* and PBS control groups after systemic administration of the bacteria or PBS. And the haematological examination and H&E staining results demonstrated that systemic administration of the GVs@*E. coli* combined with acoustic manipulation did not cause any apparent pathological damage to peripheral blood, liver functions and major organs.

As can be known from our experimental results and references [2], injecting bacteria of magnitude 10^7 does not generally cause significant toxic reactions to the organism, but we still recommend the use of non-pathogenic bacteria or to inject less bacteria while ensuring control and therapeutic efficacy.

Reference:

[1] Fonseca, A. D. C., Kohler, T., & Ahmed, D. (2022). Ultrasound - Controlled Swarmbots Under Physiological Flow Conditions. *Advanced Materials Interfaces*, 9(26), 2200877.

[2] Yi, X., Zhou, H., Chao, Y., Xiong, S., Zhong, J., Chai, Z., ... & Liu, Z. (2020). Bacteria-triggered tumour-specific thrombosis to enable potent photothermal immunotherapy of cancer. *Science advances*, 6(33), eaba3546.

6) Can the authors explain the maximum cluster size? Also, will the bacteria cluster block the blood flow? Can the authors give some insight?

Response:

Thanks for reviewer's helpful question. As the force analysis shown in Fig. 2i in the manuscript, from the range of -0.66mm to 0.66mm, the direction of the primary acoustic radiation force is toward the focal center, regardless of whether the bacteria are to the left or right of the focus center. Beyond this range, the bacteria will begin to be repulsed away from the center. Therefore, the size of the largest cluster would not exceed 1.32 mm under the conditions of sufficient bacterial quantity and aggregation time. Under our experimental conditions, the maximum cluster size was observed to be about 830 μm . In our study, although the size of the cluster may be larger than the vessel diameter, it is unlikely for the bacteria clusters to block the blood flow, except in very thin vessels with very low blood flow velocity. As explained in the answer of question 2, the process of manipulating tiny bacteria in the tube to form clusters is a process in which particles attract each other in the transverse direction and

simultaneously are migrated towards the tube wall along the wave propagation direction. Eventually, the bacteria are pushed and trapped against the wall of the tube and gather laterally toward the center of the acoustic trap to form clusters. The cluster we observed was clung to the tube wall and its projection is elliptical. However, the thickness of the cluster in the vertical projection direction is finite, so it is difficult to fill the entire cavity.

7) Will these trapped *E.coli* escape from the focus point? What is the minimum trapping force require to trap and manipulate bacteria cluster?

Response:

Thanks for reviewer's insightful question. When the GVs@*E. coli* have gathered to form clusters, it is difficult for bacteria to escape from the clusters due to the close binding under the considerable secondary radiation force. The escape of bacteria from the focus point may occur in the stage where the bacteria are driven towards the focus center under the primary acoustic radiation force and do not form clusters yet. In this process, bacteria are mainly affected by primary acoustic radiation force and Stokes force. The primary radiant forces will promote the GVs@*E. coli* to move towards the focal center, while Stokes force will impede this process. As discussed in the answer of question 4, bacteria in this case are in a process of first acceleration and then uniform movement. Thus, Stokes force, which depends mainly on dynamic viscosity, radius and velocity of the bacterium, also first increases and then reach a constant value (called critical Stokes force, F_s). Thus, when the bacterium is moved near the

focus center, the minimum primary radiant force exerted on the bacterium without significant secondary radiant force must be greater than F_s , otherwise the bacterium will be driven following the acoustic streaming and possible to escape from the focus. When individual bacterium gather together to form a cluster, the cluster can be regarded as a whole target. The minimum radiation force also need to be greater than the corresponding critical Stokes force F_s defined by the radius and velocity of the cluster. But at a size of a few hundred microns of the cluster, the radiation force is generally much greater than the Stokes force because the radiant force is proportional to the radius of the target, while the Stokes force is proportional to the radius of the target. This makes it easier to manipulate clusters than individual bacterium.

8) “When the preset topological charge m changes from 1 to 3” what is changed? Parameters, phased configuration, or amplitude? What is the difference?

Response:

Thanks for reviewer’s question. Topological charge m of the vortex demonstrates the vortex phase changes its value in m cycles of 2π in any closed circuit about the vortex axis. And the magnitude and sign of m define the speed of phase change and the direction of the vortex. In this study, we applied the holographic acoustic element framework method (HAEFM) for calculating the amplitude and phase information of the element of the array. In this method, the IB algorithm is applied to achieve the focusing phase, and a spiral-patterned holographic signature with a changeable topological charge m is used to create the vortex. The spiral-patterned holographic

signature can be calculated by the equation $\varphi=m\cdot\theta$, where φ is the signature phase and θ is the polar angle of each element of the array relative to the center of the vortex. When m changes from 1 to 3, the signature phase φ and the total phase of each element, which is equal to focusing phase plus the signature phase, changes accordingly. By using the total phase information to excite the array, the vortex acoustic field corresponding to the topological charge can be formed.

9) “the rotation velocity decreases with increasing topological charge” why?

Response:

Thanks for reviewer’s question. Acoustic vortex carry orbital angular momentum (OAM) owing to its helical wavefront. A radiation torque exerted on the objects in the acoustic vortex field was accompanied by energy transfer owing to the transfer of OAM. When a small particle manipulated by vortex waves is not aligned with the vortex core, the particle will rotate in an orbit centered on the vortex core under the action of the radiation torque. When the radiation torque was balance with the viscous-drag torque on the rotated particle in the fluid, the steady rotation rate will be acquired, which is proportional to the intensity of the vortex field [1]. When the topological charge m of the vortex field changes from 1 to 3, the distances between the strongest intensity peaks in the vortex fields increase accordingly (approximately 0.9, 1.4, and 1.8 mm, respectively). Thus the area of the vortex intensity ring is constantly increasing, causing a more dispersed intensity distribution. When the same voltage is used to excite the array, the total acoustic power in the space (mainly

concentrated in the vortex intensity ring) is the same, which can be expressed as the integral of sound intensity and area of normal surface. As the total acoustic power of the intensity ring is the same, when m changes from 1 to 3, the area of the vortex intensity ring becomes larger, so the corresponding average acoustic intensity will decrease. This result has been proved by our simulation data that the normalized maximum acoustic intensity of the vortex acoustic field ($m=1, 2, 3$) is about 1, 0.60, 0.41, respectively. Further, the rotation velocity of the cluster will decrease with increasing topological charge owing to the decreasing acoustic intensity.

Reference:

[1] Zhang, L. (2018). Reversals of orbital angular momentum transfer and radiation torque. *Physical Review Applied*, 10(3), 034039.

10) “shows promise for application to other therapeutic cells such as CAR-T cells, natural killer cells, or stem cells in future clinical applications” ? How are GVs generated in these cells?

Response:

Thanks for reviewer’s question. In fact, it has been reported in the literature that a eukaryotic genetic program is engineered by Mikhail G. Shapiro et al, whose introduction into mammalian cells (human embryonic kidney (HEK) 293T cells) results in the expression of GVs [1]. This eukaryotic genetic program is called mammalian acoustic reporter genes (mARGs), just starting with a gene cluster derived from bacteria and realizing the improvement of the expression of acoustic reporter

genes in prokaryotes to in eukaryotes. The mARGs is synthesized by individual gas vesicle genes from three different microbial species using codons optimized for human expression, and cloned each gene into a separate monocistronic plasmid, and then transiently co-transfected mixtures of the genes from each species into mammalian cells. The study establish the ability of an engineered genetic construct encoding prokaryote-derived GVs to serve as a mammalian reporter gene for ultrasound, and this genetic constructs is feasible for other human mammalian cells, such as CAR-T cells, natural killer cells, or stem cells.

Reference:

[1] Farhadi, A., Ho, G. H., Sawyer, D. P., Bourdeau, R. W., & Shapiro, M. G. (2019). Ultrasound imaging of gene expression in mammalian cells. *Science*, 365(6460), 1469-1475.

11) What is the acoustic attenuation through air, skin, tissue, and vessel?

Response:

Thanks for reviewer's question. Acoustic attenuation refers to the phenomenon that the intensity of sound waves decreases gradually with the increase of the propagation distance when they are propagated in the media. The acoustic attenuation in most biological tissues over the MHz frequency range has been proved to be to obey the law of exponential attenuation, which can be expressed by the following equation [1],

$$I = I_0 e^{-2\alpha x} \quad (2)$$

Where I and I_0 represent the acoustic intensity after attenuation and incident acoustic intensity, x is propagation distance, and α is the attenuation coefficient of the medium.

When discussing attenuation related to the properties of the medium, acoustic attenuation can be divided into absorption attenuation and scattering attenuation.

When the medium is air, scattering attenuation is dominant, and the attenuation coefficient through air is about 6.3 dB/cm (25 °C, 1 atm, 2 MHz) [2]. For skin, tissue, and vessel, absorption attenuation is dominant, which is proportional to the frequency.

The attenuation coefficient through soft tissue (such as skin, tissue, and vessel), blood, fat and muscle is about 1, 0.18, 0.68, 1.3 dB/(cm • MHz), respectively [3]. In this study, mouse skin-fold dorsal window chamber model and tumour site are both belong to soft tissue, and their thicknesses are very thin, so the acoustic attenuation is not obvious. And the gaps between array and skin are filled with ultrasound coupling agent to avoid scattering of air.

Reference:

[1] Feng, R., Yao, J., & Guan, L. (1999). Ultrasonics handbook. Publishing Company of Nanjing University, Nanjing.

[2] Kong Tao, Xu Chunguang, Zhang Yuntao,&Xu Hanhui (2011). Calculation and measurement of sound field of air coupled ultrasonic transducer Journal of Mechanical Engineering, 47 (22), 19-24.

[3] Zhu, J. (2017). Ultrasonic principle and biomedical engineering application: biomedical ultrasonics. Publishing Company of Shanghai Jiaotong University, Shanghai.

12) Does the transducer array touch the body? What is the thermal effect?

Response:

Thanks for reviewer's question. In our experiment, the surface of the array did not directly touch the skin of the mice. The focal length of the array was set at 6 mm, and the array transducer was positioned on the mouse skin-fold dorsal window chamber model or tumour site. The gaps were filled with ultrasound coupling agent. In addition, we also monitored the process of ultrasound therapy using a thermal imaging instrument, and found no significant temperature rise in skin and tissue received with acoustic irradiation during 20 min.

13) In fig. 2g and 2h, how the force direction is defined? Why a negative contrast factor resulted in the force directed toward the focus?

Response:

Thanks for reviewer's question. In Fig. 2g and 2h, we have defined a rectangular coordinate system with the center of the focal acoustic field as the origin, and the two orthogonal directions as the x and y axes. In this coordinate system, grid points are selected from x axis -0.5 mm to 0.5 mm and y axis -0.5 mm to 0.5 mm in the x and y directions with a spacing of 0.1 mm respectively. The acoustic radiant force exerted at each grid point is calculated to represent the direction and magnitude of the radiant force at the corresponding location in the focal acoustic field. If the force component of the acoustic radiation force on the x axis is in the same direction as the positive

direction of the x axis, the radiation force exerted at the grid point in the x direction is positive; otherwise, it is negative. The situation in y axis case is similar. On the basis of the establishment of coordinate system and the selection of grid points, we use the gradient of Gorkov potential to calculate the radiation force of cells whose diameters are much smaller than the wave length in the acoustic field (see the Methods section in the manuscript for detail). The corresponding radiant force can be obtained by substituting the acoustic pressure at the grid points and the cell size, density, compressibility and other parameters.

The primary acoustic radiant force applied to the cells depends on several factors, including the acoustic pressure amplitude, cell volume, wavelength, and acoustic contrast factor. Especially, the sign of the acoustic contrast factor, which depends on both the density and the compressibility of the cell relative to the surrounding solution, dictates the direction of the radiant force exerted on the cell. For the GVs@*E. coli* which has a negative contrast factor (cells with higher compressibility and lower density than the surrounding media), the direction of the radiant force exerted on the GVs@*E. coli* points from the weak-pressure-amplitude position to the strong-pressure-amplitude position [1][2]. Therefore, GVs@*E. coli* near the focal zone will converge towards the center of the focus, where the pressure of the acoustic field is strongest. This result can also be seen from Fig. 2i, where the radiation force exerted on *E. coli* and GVs@*E. coli* by the focal beam in the x-axis is calculated. On the left side of the focus, a positive acoustic radiation force that pointed to the right was exerted on the GVs@*E. coli*, while on the right side it was the opposite. So the

radiation force exerted on GVs@*E. coli* which are on both sides of the focus points towards the center of focus.

Reference:

[1] Johnson, L. M., Gao, L., Shields IV, C. W., Smith, M., Efimenko, K., Cushing, K., ... & López, G. P. (2013). Elastomeric microparticles for acoustic mediated bioseparations. *Journal of nanobiotechnology*, 11(1), 1-8.

[2] Yang, Y., Ma, T., Li, S., Zhang, Q., Huang, J., Liu, Y., ... & Zheng, H. (2021). Self-navigated 3d acoustic tweezers in complex media based on time reversal. *Research*, 2021.

14) There is almost no references in the manuscript regarding acoustic manipulation of gas based vesicles. This is the main principle for the manipulation of bacteria, and it' s not the first time that has been done. Please cite the following articles *in vivo* acoustic manipulation of microparticles in zebrafish embryos , *Sci. Adv.* 2022, 8, eabm2785.

Response:

Thanks for reviewer's constructive comments. We have added the reviewer suggested *Reference 38*: Jooss, V. M., Bolten, J. S., Huwyler, J., & Ahmed, D. (2022). In vivo acoustic manipulation of microparticles in zebrafish embryos. *Science advances*, 8(12), eabm2785.

The manuscript has been revised as follows in P5, lines 80-81:

P5, lines 80-81: “Some research has been published on *in-vivo* manipulation of non-living objects, such as the manipulation of glass spheres in a pig bladder³⁶, the trapping of microbubbles in mouse back epidermal blood vessels³⁷ and manipulation of microparticles in zebrafish embryos³⁸.”.

Response:

Some other revisions have also been made in this manuscript:

1. We have adjusted the serial number of the Supplementary Figures.
2. We have modified the Supplementary Movie S2, which mainly replacing the parts of trapping bacteria cluster in 90- and 120- μm vessels in the original video.
3. We adjusted the order of authors and changed Yuanyuan Wang from the seventh author to the fourth author. Moreover, we have added another affiliated institution of Yaozhang Yang, which is Shenzhen Bay Laboratory.
4. We have added to the acknowledgments for Mr. Xin Xu, Ms. Ying Zhou, Ms. Bing Wang and added another founding, which is the China Postdoctoral Science Foundation (2022M723305) in Acknowledgements section.
5. Some of the formats have been adjusted for consistency and some words and grammar mistakes have been corrected.

REVIEWER COMMENTS

Reviewer #1 (Remarks to the Author):

My comments on the original manuscript were minor and I am happy that they have been adequately addressed.

Reviewer #2 (Remarks to the Author):

1. Experimental results:

The new tumor treatment results obtained using GV-expressing Salmonella cells (YB1) are a good addition to the manuscript. A few comments on this new data:

I think it is important to include data demonstrating GV expression specifically in YB1 cells since these are the cells used in the most critical in vivo experiment. The experiments can be similar to Fig 2, b-e and Sup Fig 1. YB1 cells are anaerobic, and I could not find reports of GVs previously being expressed in anaerobes.

I am also confused by the statement in the Methods that YB1 cells were cultured the same way as E. Coli. This is not what I would expect for anaerobes. Could the authors please clarify/comment?

In addition, since YB1 cells are presumably capable of flagellar motion, it would be useful to demonstrate how well they get trapped at the acoustic focus compared to non-motile E. Coli BL21. I think quantifying this is important because the tumor growth improvement provided by applying ultrasound is fairly modest compared to non-US GVs@YB1 controls (Fig 7b), and it would be helpful to understand why it is not larger (i.e. where is there room for improvement in future studies).

In addition, the tumor therapy experiments in Fig. 7b seem to be missing a control: the combination of ultrasound with YB1 cells that do not express GVs (i.e. YB1+US). Since focused ultrasound can have some effects on its own (as seen in previous studies of tumor models where thermal or mechanical effects led to retarded tumor growth), and YB1 cells can have effects on their own, one might expect additive or synergistic effects between them. Given the absence of direct evidence for GV expression in YB1 cells in the revised manuscript, it is reasonable to worry that the modest difference seen between GVs@YB1 and GVs@YB1+US could be not due to GVs, but due to the YB1+US combination.

In the colony counts in Supp. Fig 8 the difference seen at 7 days between with and without ultrasound looks very small. What is the fold-difference? (This could be stated above the bars or in the figure caption). What is the justification for only looking at/after 7 days post-administration and not earlier? Wouldn't the effect of US-based concentration enhancement be expected to be strongest immediately after treatment (after allowing time for washout from vasculature or flushing it), when a much smaller number of cells makes it into the tumor? By 7 or 14 days, the bacterial population in the tumor will have presumably undergone many cell divisions and might already reach an equilibrium density set by the tumor niche and/or be subject to population change resulting from tumor growth. Perhaps an earlier time point would more directly demonstrate the core concept of this in vivo demonstration that acoustic trapping increases the initial migration of GV-expressing bacteria into a tumor.

By the way, the plate examples for colony counts in Sup Fig 8 look quite dense. Are satellite colonies a concern at this density?

The Methods say that the tumors in these experiments were treated with 20 min of 10% DC

ultrasound at 20V. What peak negative pressure does this correspond to? This information is useful to compare this treatment to previous studies on focused ultrasound treatment of tumors and uses of GV-expressing bacteria as tumor-damaging cavitation agents.

2. Comments on the text:

Some early sections of the manuscript still read like they are describing an unprecedented result for the very first time when in fact they are reproducing published findings. I think this can be clarified with a few small tweaks. For example, the authors could consider (edits indicated with asterisks):

Abstract: "In this study, genetically engineered bacteria that can produce numerous sub-micron gas vesicles (GVs) in the bacterial cytoplasm are **generated** by heterologous expression of gene clusters encoding GV. The presence of GV significantly enhance the acoustic sensitivity of the genetically engineered bacteria (called GV@E. coli), **which has been shown to** enable them to be manipulated by ultrasound."

Main text:

"In details, we genetically engineered E. coli through the heterologous expression of gene clusters encoding GV, which significantly enhanced the acoustic contrast of these bacteria relative to water (Fig. 1, the acoustic contrast has increased by nearly 15 times), **consistent with previous results (45)**"

"To **validate** the of acoustic manipulation of bacteria **in our system**, we firstly examined the trapping capabilities of genetically engineered GV@E. coli and uninduced control E. coli by using the acoustic tweezers."

"This strong ARF with the direction pointing to the focal center will help these genetically engineered bacteria with GV to overcome their own Stokes force and will drive them to aggregate in the focal area **(45)**." (The clustering of bacteria at a focal point was also demonstrated in ref 45.)

Reviewer #3 (Remarks to the Author):

The authors did a great job! I recommend the manuscript for publication.

Reviewers' comments:

Reviewer #1 (Remarks to the Author):

My comments on the original manuscript were minor and I am happy that they have been adequately addressed.

Response:

Thanks again for reviewer's insightful comments. The manuscript has been significantly improved owing to the reviewer's suggestions. Wish the reviewer all the best.

Reviewer #2 (Remarks to the Author):

1. Experimental results:

The new tumor treatment results obtained using GV-expressing Salmonella cells (YB1) are a good addition to the manuscript. A few comments on this new data:

I think it is important to include data demonstrating GV expression specifically in YB1 cells since these are the cells used in the most critical in vivo experiment. The experiments can be similar to Fig 2, b-e and Sup Fig 1. YB1 cells are anaerobic, and I could not find reports of GVs previously being expressed in anaerobes.

Response:

Thanks a lot for reviewer's constructive comments. According to the previous studies, we found that the *S. typhimurium* ELH1301 could express GVs [1] and YB1 had an effective function of tumor treatment [2]. So, we attempted to use YB1 to express GVs. And similarly experiments to verify the feasibility of YB1 expressing GVs were

also carried out. The induced GV@YB1 was centrifuged (350 g, 3 h), and the GV@YB1 on the upper layer of the solution was collected. After standing for one day, GV@YB1 appeared to float up due to the effect of GVs buoyancy (Figure 1a). Meanwhile, the result of TEM showed that the cytoplasm of GV@YB1 was filled with a large amount of GVs (Figure 1b).

We have added Figure 1 to the supplementary information as Supplementary Fig 8.

Figure 1. GVs expression of GV@YB1. **a**, GV@YB1 were collected into PBS solution and GV@YB1 floated to the top of the solution after standing for 24 h (white). **b**, Transmission electron microscope image of GV@YB1. GVs were filled with the cytoplasm.

The manuscript has been revised as follows in P15, Lines 298-301:

“The GV expression in YB1 (Supplementary Fig. 8a-8b) and the acoustic manipulation of GV@YB1 (Supplementary Fig. 9a-9c) were firstly demonstrated. The acoustic trapping of GV@YB1 and GV@*E. coli* showed similar trends and it seemed to take longer to trap GV@YB1 mainly because YB1 had flagellar motion.”

References

- [1] Bourdeau, R. W., Lee-Gosselin, A., Lakshmanan, A., Farhadi, A., Kumar, S. R., Nety, S. P., & Shapiro, M. G. (2018). Acoustic reporter genes for noninvasive imaging of microorganisms in mammalian hosts. *Nature*, 553(7686), 86-90.

[2] Chen, F., Zang, Z., Chen, Z., Cui, L., Chang, Z., Ma, A., ... & Cai, L. (2019). Nanophotosensitizer-engineered Salmonella bacteria with hypoxia targeting and photothermal-assisted mutual bioaccumulation for solid tumor therapy. *Biomaterials*, 214, 119226.

I am also confused by the statement in the Methods that YB1 cells were cultured the same way as *E. Coli*. This is not what I would expect for anaerobes. Could the authors please clarify/comment?

Response:

Thanks very much for reviewer's insightful comments. We apologized for the lack of clarity in the manuscript. There were some differences of the culture mode between YB1 cells and *E. coli*, including the diaminopimelic acid (DAP) with a final concentration of 50 µg/ml was added to the LB liquid medium to ensure the normal growth of YB1 [1].

The manuscript has been revised as follows in Methods (P19 Lines 383-386):

“The culture process of transformed YB1 was similar with *E. coli* BL21, except the diaminopimelic acid (DAP, sigma #D1377) with a final concentration of 50 µg/ml was added to the LB liquid medium, ...”

References:

[1] Yu, B., Yang, M., Shi, L., Yao, Y., Jiang, Q., Li, X., ... & Huang, J. D. (2012). Explicit hypoxia targeting with tumor suppression by creating an “obligate” anaerobic Salmonella Typhimurium strain. *Scientific reports*, 2(1), 1-10.

In addition, since YB1 cells are presumably capable of flagellar motion, it would be useful to demonstrate how well they get trapped at the acoustic focus compared to non-motile *E. Coli* BL21. I think quantifying this is important because the tumor growth improvement provided by applying ultrasound is fairly modest compared to non-US GV@YB1 controls (Fig 7b), and it would be helpful to understand why it is not larger (i.e. where is there room for improvement in future studies).

Response:

Thanks a lot for reviewer's constructive comments. We supplemented the relevant experiments of acoustic trapping of GV@YB1 and YB1 cells in a silicone tube, and the results showed only GV@YB1 could be trapped to form a cluster by the acoustic radiation force while the YB1 cells would be washed away by the acoustic streaming (Figure 2a-2b). And we have quantified the acoustic trapping experiments under different voltage (10V, 15V, 25V, Figure 2c), duty cycle (8%, 10%, 12%, Figure 2d), and concentration (100, 200, 400, unit: $\times 10^7/\text{ml}$, Figure 2e). Under these conditions, acoustic trapping of GV@YB1 and GV@*E. coli* showed similar trends. And under the same conditions (input voltage: 15V, duty cycle: 10%), the GV@*E. coli* could aggregate to form a 150 μm cluster at 15 s, while GV@YB1 could form a 100 μm cluster at 22 s. This difference can be mainly considered as the effect of flagellar action of GV@YB1. Furthermore, we used electronically steered focal beams to trap and drive these genetically engineered GV@YB1 along a programmable trajectory in a silicone tube (Figure 2f).

We have added Figure 2 to the supplementary information as Supplementary Fig 9.

Figure 2. Acoustic trapping and manipulation of GV@YB1. a, b, Microscopic fluorescence photographs of control YB1 (a) and GV@YB1 (b) in silicone tube being exposed to ultrasound (focal beam) for 30 s, respectively. Only GV@YB1 can aggregate at the focal beam centre and form clusters under the dominance of the ARF. c, d, Bacterial cluster formation times (100 μm cluster diameter) and cluster sizes (after ultrasound had been turned on for 2 min) of the GV@YB1 under different input voltages (c) and duty cycles (d). ** means $P < 0.01$ compared to the 10 V group in (c). e, Bacterial cluster size of the GV@YB1 with various concentrations under the same ultrasound conditions. * means $P < 0.05$ and **** means $P < 0.0001$ compared to the $100 \times 10^7/\text{ml}$ case in (e). f, Manipulating the GV@YB1 cluster along a straight line within the range of ± 0.6 mm by electronically steering the focal beams. Data in (c), (d), and (e) are presented as mean \pm s.d. ($n = 3$).

The manuscript has been revised as follows in P15, Lines 298-301:

“The GV expression in YB1 (Supplementary Fig. 8a-8b) and the acoustic manipulation of GV@YB1 (Supplementary Fig. 9a-9c) were firstly demonstrated. The acoustic trapping of GV@YB1 and GV@*E. coli* showed similar trends and it seemed to take longer to trap GV@YB1 mainly because YB1 had flagellar motion.”

In addition, the tumor therapy experiments in Fig. 7b seem to be missing a control: the combination of ultrasound with YB1 cells that do not express GVs (i.e. YB1+US).

Since focused ultrasound can have some effects on its own (as seen in previous

studies of tumor models where thermal or mechanical effects led to retarded tumor growth), and YB1 cells can have effects on their own, one might expect additive or synergistic effects between them. Given the absence of direct evidence for GV expression in YB1 cells in the revised manuscript, it is reasonable to worry that the modest difference seen between GV@YB1 and GV@YB1+US could be not due to GVs, but due to the YB1+US combination.

Response:

Thanks very much for reviewer's insightful comments. In this study, we aimed to verify the ability of acoustic manipulation of GV@*E.coli* or GV@YB1 and further investigate the enhancement of ultrasound manipulation in the tumor therapy, which can trap more GV@YB1 in the tumor. To illustrate this point, we firstly supplemented the evidences that YB1 cells express GVs this time. We performed the experiments of observing the float of GV@YB1 (Figure 1a), TME on GV@YB1 (Figure 1b).

Figure 1. GV expression of GV@YB1. a, GV@YB1 was collected into PBS solution and after the solution stand, GV@YB1 floated to the top of the solution (white). **b,** Transmission electron microscope image of GV@YB1, GVs filled with the cytoplasm.

Furthermore, we supplemented the relevant experiments of acoustic trapping of GVs@YB1 and YB1 cells in a silicone tube, and the results showed only GVs@YB1 could be trapped to form a cluster by the acoustic radiation force while the YB1 cells would be washed away by the acoustic streaming (Figure 2a-2b). Finally, we supplemented the experiment by counting the number of GVs@YB1 in the tumor on the first day after ultrasound irradiation. The results showed that the number of bacterial colonies in the GVs@YB1+US group was higher than that in the GVs@YB1 group (Figure 3a) and the fold-difference between GVs@YB1 + US and GVs@YB1 reached 4.5 (Figure 3b), indicating that more GV-expressing bacteria were trapped in the tumor under the effect of ultrasound.

Figure 2. Acoustic trapping and manipulation of GVs@YB1. **a, b,** Microscopic fluorescence photographs of control YB1 (**a**) and GVs@YB1 (**b**) in silicone tube being exposed to ultrasound (focal beam) for 30 s, respectively. Only GVs@YB1 can aggregate at the focal beam centre and form clusters under the dominance of the ARF. **c, d,** Bacterial cluster formation times (100 μm cluster diameter) and cluster sizes (after ultrasound had been turned on for 2 min) of the GVs@YB1 under different input voltages (**c**) and duty cycles (**d**). ** means $P < 0.01$ compared to the 10 V group in (**c**). **e,** Bacterial cluster size of the GVs@YB1 with various concentrations under the same ultrasound conditions. * means $P < 0.05$ and **** means $P < 0.0001$ compared to the $100 \times 10^7/\text{ml}$ case in (**e**). **f,** Manipulating the GVs@YB1 cluster along a straight line within the range of ± 0.6 mm by electronically steering the focal beams. Data in (**c**), (**d**), and (**e**) are presented as mean \pm s.d. ($n = 3$).

Figure 3. Acoustic aggregation of GV@YB1 in tumour. **a.** The number of GV@YB1 in the tumour of tumour-bearing mice with or without ultrasonic irradiation, at 1 day after intravenous administration of 5×10^7 CFU GV@YB1. **b.** The Quantification of (a), in the 1 day, the fold-difference between GV@YB1 + US and GV@YB1 reached 4.5. **** means $P < 0.0001$.

By referring to some similar researches [1-3], we set two groups of GV@YB1 and GV@YB1+US respectively in tumor therapy to demonstrate the enhancement of ultrasound manipulation. Herein, the result of tumor therapy showed that the US group alone did not significantly inhibit tumor growth compared with the control group (Fig. 7b, $P > 0.1$). The YB1 group alone seemed to have limited effect for tumor treatment, which was similar to the alone GV@YB1 group. And our supplementary experiments had also proved that ultrasound could only aggregate GV-expressing YB1 (GV@YB1) into clusters but not alone YB1 group. Therefore, these results can show that the difference seen between GV@YB1 and GV@YB1+US in the tumor treatment were due to GVs, but not due to the YB1+US combination.

References:

- [1] Yue, W., Chen, L., Yu, L., Zhou, B., Yin, H., Ren, W., ... & Chen, Y. (2019). Checkpoint blockade and nanosensitizer-augmented noninvasive sonodynamic therapy combination reduces tumour growth and metastases in mice. *Nature communications*, 10(1), 2025.

[2] Zheng, P., Fan, M., Liu, H., Zhang, Y., Dai, X., Li, H., ... & Li, Z. (2020). Self-propelled and near-infrared-phototoxic photosynthetic bacteria as photothermal agents for hypoxia-targeted cancer therapy. *ACS nano*, 15(1), 1100-1110.

[3] Zheng, D. W., Chen, Y., Li, Z. H., Xu, L., Li, C. X., Li, B., ... & Zhang, X. Z. (2018). Optically-controlled bacterial metabolite for cancer therapy. *Nature Communications*, 9(1), 1680.

In the colony counts in Supp. Fig 8 the difference seen at 7 days between with and without ultrasound looks very small. What is the fold-difference? (This could be stated above the bars or in the figure caption). What is the justification for only looking at/after 7 days post-administration and not earlier? Wouldn't the effect of US-based concentration enhancement be expected to be strongest immediately after treatment (after allowing time for washout from vasculature or flushing it), when a much smaller number of cells makes it into the tumor? By 7 or 14 days, the bacterial population in the tumor will have presumably undergone many cell divisions and might already reach an equilibrium density set by the tumor niche and/or be subject to population change resulting from tumor growth. Perhaps an earlier time point would more directly demonstrate the core concept of this in vivo demonstration that acoustic trapping increases the initial migration of GV-expressing bacteria into a tumor.

Response:

Thanks a lot for reviewer's constructive comments. In the supplementary information, we have added the fold-difference (1.29 in 7 days and 2.67 in 14 days) in the figure caption of Supplementary Fig. 8 (Supplementary Fig. 10 in updated version). In Fig. 6a in the manuscript, the 0, 2, and 4 min fluorescence imaging of tumour-bearing mice after injection of GV@*E. coli* with or without acoustic manipulation had been shown, indicating the ultrasound could trap more GV-expressing bacterial immediately after treatment. In addition, we supplemented the experiment by counting the number of GV@YB1 in the tumor on the first day after ultrasound irradiation. The results showed that the number of bacterial colonies in the GV@YB1+US group was higher than that in the GV@YB1 group (Figure 3a) and the fold-difference between GV@YB1 + US and GV@YB1 reached 4.5 (Figure 3b), indicating that more bacteria were still trapped in the tumor under the effect of ultrasound 1 day after the treatment.

We have added Figure 3 to the supplementary information as Supplementary Fig 10.

Figure 3. Acoustic aggregation of GV@YB1 in tumour. **a.** The number of GV@YB1 in the tumour of tumour-bearing mice with or without ultrasonic irradiation, at 1 day after intravenous administration of 5×10^7 CFU GV@YB1. **b.** The Quantification of (a), in the 1 day, the fold-difference between GV@YB1 + US and GV@YB1 reached 4.5. **** means $P < 0.0001$.

The manuscript has been revised as follows in the figure caption of updated Supplementary Fig. 10 (P11, Lines 83-85):

“... b, Quantification of (a), in the 7 days, the fold-difference between GV_s@YB1 + US and GV_s@YB1 reached 1.29, and 2.67 in 14 days.”

The manuscript has been revised as follows (P16, Lines 315-317):

“Furthermore, we detected the GV_s@YB1 aggregation in the tumour of tumour-bearing mice at day 1, day 7 and day 14 after treatment. Our results showed that the number of GV_s@YB1 in the tumour of GV_s@YB1 + US group was significantly higher than that in GV_s@YB1 group (Supplementary Fig. 10a-10b and Supplementary Fig. 11a-11b).”

By the way, the plate examples for colony counts in Sup Fig 8 look quite dense. Are satellite colonies a concern at this density?

Response:

Thanks very much for reviewer’s insightful comments. When the bacterial growth density is too high, it can lead to the generation of satellite colonies, which has an impact on the statistics of bacterial population. Therefore, when using the ImageJ software for statistics, the statistical condition of “size=1-intensity” was set in order to exclude satellite colonies.

The manuscript has been revised as follows in Method (P24, Lines 489-491) to illustrate this:

“..., which was counted by the ImageJ software, and statistical condition of “size=1-intensity” was set in order to exclude satellite colonies.”

The Methods say that the tumors in these experiments were treated with 20 min of 10% DC ultrasound at 20V. What peak negative pressure does this correspond to? This information is useful to compare this treatment to previous studies on focused ultrasound treatment of tumors and uses of GV-expressing bacteria as tumor-damaging cavitation agents.

Response:

Thanks a lot for reviewer’s constructive comments. According to the measurement results, the peak negative pressure (PNP) in the focal spot is about 1.11 MPa at 20V input voltage. In addition, the center frequency of the ultrasound wave is 3 MHz, thus the *MI* value is about 0.64, which is also much less than the corresponding U.S. Food and Drug Administration (FDA) restrictions for acoustic output from diagnostic ultrasound equipment ($MI \leq 1.9$ [1]). In addition, according to the reference [2] (Fig. 4d in the reference), the GV-expressing *S. typhimurium* cells began to emit a high level of broadband signals in response to ultrasound pulses when $PNP \geq 0.76$ MPa at 300 kHz, indicating efficient collapse and subsequent cavitation occurred. And in this case, the corresponding *MI* value is approximately 1.39, which is more than twice the *MI* in this study. Therefore, under the input energy conditions in this study, it is difficult to cause the GVs in the GV-expressing bacteria collapse and further cavitation.

Reference:

[1] Nelson, T. R., Fowlkes, J. B., Abramowicz, J. S., & Church, C. C. (2009).

Ultrasound biosafety considerations for the practicing sonographer and sonologist.

[2] Bar-Zion, A., Nourmahnad, A., Mittelstein, D. R., Shivaiei, S., Yoo, S., Buss, M.

T., ... & Shapiro, M. G. (2021). Acoustically triggered mechanotherapy using genetically encoded gas vesicles. *Nature nanotechnology*, 16(12), 1403-1412.

2. Comments on the text:

Some early sections of the manuscript still read like they are describing an unprecedented result for the very first time when in fact they are reproducing published findings. I think this can be clarified with a few small tweaks. For example, the authors could consider (edits indicated with asterisks):

Response:

Thanks a lot for reviewer's insightful comments. We agree with you that our study has been inspired by Wu., et al, who have already introduced the concept of GV-based acoustic manipulation and demonstrated that GV-expressing *E. coli* can be selectively manipulated by ultrasound *in vitro* [1]. Therefore, we will revise our manuscript in good faith in accordance with your suggestions.

Reference:

[1] Wu, D., Baresch, D., Cook, C., Malounda, D., Maresca, D., Abundo, M. P., ... &

Shapiro, M. G. (2019). Genetically encoded nanostructures enable acoustic manipulation of engineered cells. *bioRxiv*, 691105.

Abstract: “In this study, genetically engineered bacteria that can produce numerous sub-micron gas vesicles (GVs) in the bacterial cytoplasm are *generated* by heterologous expression of gene clusters encoding GVs. The presence of GVs significantly enhance the acoustic sensitivity of the genetically engineered bacteria (called GVs@E. coli), *which has been shown to* enable them to be manipulated by ultrasound.”

The abstract has been revised as follows in P2, Lines 22 and 24:

“In this study, genetically engineered bacteria that can produce numerous sub-micron gas vesicles (GVs) in the bacterial cytoplasm are **generated** by heterologous expression of gene clusters encoding GVs. The presence of GVs significantly enhance the acoustic sensitivity of the genetically engineered bacteria (called GVs@E. coli), **which has been shown to enable** them to be manipulated by ultrasound.”

Main text:

“In details, we genetically engineered E. coli through the heterologous expression of gene clusters encoding GVs, which significantly enhanced the acoustic contrast of these bacteria relative to water (Fig. 1, the acoustic contrast has increased by nearly 15 times), *consistent with previous results (45)*”

The manuscript has been revised as follows in P5, Lines 99-100:

“..., which significantly enhanced the acoustic contrast of these bacteria relative to water (Fig. 1, the acoustic contrast has increased by nearly 15 times), consistent with previous results⁴⁵”

“To *validate* the of acoustic manipulation of bacteria *in our system*, we firstly examined the trapping capabilities of genetically engineered GV@*E. coli* and uninduced control *E. coli* by using the acoustic tweezers.”

The manuscript has been revised as follows in P7, Lines 133-134:

“To validate the acoustic manipulation of bacteria in our system, we firstly examined the trapping capabilities of genetically engineered GV@*E. coli* and uninduced control *E. coli* by using the acoustic tweezers.”

“ This strong ARF with the direction pointing to the focal center will help these genetically engineered bacteria with GVs to overcome their own Stokes force and will drive them to aggregate in the focal area *(45)*. ” (The clustering of bacteria at a focal point was also demonstrated in ref 45.)

The manuscript has been revised as follows in P8, Line 149:

“This strong ARF with the direction pointing to the focal center will help these genetically engineered bacteria with GVs to overcome their own Stokes force and will drive them to aggregate in the focal area⁴⁵.”

Reviewer #3 (Remarks to the Author):

The authors did a great job! I recommend the manuscript for publication.

Response:

Thanks very much for reviewer's constructive comments. According to the reviewer's suggestions, the manuscript has been markedly refined. Best wishes to the reviewer.

Response:

Some other revisions have also been made in this manuscript:

1. There was a slight error in the schematic figure of Fig. 2a and we have replaced it.
2. The preprint version of reference 45 has a newer, officially published version (Wu, D., Baresch, D., Cook, C., Ma, Z., Duan, M., Malounda, D., ... & Shapiro, M. G. (2023). Biomolecular actuators for genetically selective acoustic manipulation of cells. *Science Advances*, 9(8), eadd9186.) and we have replaced it.
3. We have adjusted the serial number of the Supplementary Figures.
4. Some of the formats have been adjusted for consistency and some words and grammar mistakes have been corrected.

REVIEWERS' COMMENTS

Reviewer #2 (Remarks to the Author):

The authors have addressed my remaining concerns.